

# Explicit simulation of chemical composition, size distribution and cloud condensation nuclei of secondary organic aerosol from α-pinene ozonolysis

Zhen Song[1,2,3,4], Chenqi Zhang[1], Hongru Shen[1,†], Hao Ma[1], Iida Pullinen[6,‡], Defeng Zhao[1,3,4,5]

[1]Department of Atmospheric and Oceanic Sciences & Institute of Atmospheric Sciences, Fudan University, Shanghai, China

[2]Shanghai Frontiers Science Center of Atmosphere-Ocean Interaction, Fudan University, Shanghai, 200438, China

[3]Shanghai Key Laboratory of Ocean-land-atmosphere Boundary Dynamics and Climate Change, Fudan University, Shanghai, 200438, China

[4]National Observations and Research Station for Wetland Ecosystems of the Yangtze Estuary, Fudan University, Shanghai,

200438, China

[5]Institute of Eco-Chongming (IEC), 1050 Baozhen, Lühua Town, Chongming District, Shanghai 202151, China

[6]Institute of Energy and Climate Research, IEK-8: Troposphere, Forschungszentrum Jülich, Jülich, Germany

[†]Present address: Department of Chemical Engineering and Applied Chemistry/Department of Chemistry, University of Toronto, Toronto, Ontario M5S 3E5, Canada

[‡] Present address: Department of Applied Physics, University of Eastern Finland, Kuopio, 70210, Finland

*Correspondence to*: Defeng Zhao (dfzhao@fudan.edu.cn)

**Abstract.** Secondary organic aerosols (SOA) contribute significantly to cloud condensation nuclei (CCN), which depend on size distribution, chemical composition and hygroscopicity parameter (κ). However, how well current understanding of SOA

formation can reproduce CCN concentrations and the influence these factors on modelled CCN uncertainties are still unclear. In chemical transport models, it is difficult to address the issue due to model complexity and oversimplified representation of chemical mechanisms, particle size and κ. Here, we explicitly simulated CCN concentrations of SOA from α-pinene ozonolysis, a bench-mark system for SOA studies using a box model (PyCHAM). Using state-of-the-art treatment of chemical mechanisms, aerosol size and κ, we assessed how CCN as well as chemical composition, aerosol size and κ can be modelled against

measurement and evaluated the influence of these factors on CCN simulation. The model well simulated SOA mass concentration but overestimated O:C and H:C ratios, suggesting lack of particle-phase chemistry. Highly oxygenated molecules contributed substantially to SOA mass and thus CCN. Modeled κ closely aligned with measurements at moderate supersaturation (0.37%) but overestimate κ (by 19%) at low supersaturation (~0.19%) and underestimate κ (by 21%) at high supersaturation (0.73%). The model well reproduced particle growth, but exhibited wider and flatter size distribution compared

with measurement. The simulated CCN concentrations agreed well with measurement at moderate to high SS (0.37–0.73%) but had a significant bias at low SS. Sensitivity analysis highlights the importance of accurate representation of both size distribution and κ for CCN prediction especially at lower SS (<0.4%).

## 1 Introduction

Secondary organic aerosol (SOA), formed through the oxidation of volatile organic compounds (VOCs) and gas-particle

partitioning, constitutes a significant fraction of atmospheric submicron aerosol mass (Jimenez et al., 2009; Huang et al., 2014; Shrivastava et al., 2017). As a result, SOA contributes significantly to global cloud condensation nuclei (CCN), influencing aerosol indirect effects (aerosol-cloud interaction) and radiative forcing (IPCC, 2021).

Despite numerous research efforts, uncertainties persist in assessing indirect effects of SOA. These uncertainties are closely linked to inaccuracies in simulated CCN number concentrations within chemical transport models (IPCC, 2021). The

contribution of SOA to CCN concentrations depends on the SOA concentration (Liu and Wang, 2010; Mei et al., 2013) and CCN activity of SOA, which is determined by their particle size distribution and hygroscopicity (Farmer et al., 2015; Seinfeld



and Pandis, 2016). Therefore, uncertainties in modeled CCN levels are strongly influenced by these parameters, which are often oversimplified in current chemical transport models. For example, chemical transport models often employ simplified chemical mechanisms, such as CBM (Carbon Bond Mechanism), RACM (Regional Atmospheric Chemistry Mechanism), and

SAPRC (Statewide Air Pollution Research Center), to enhance computational efficiency and numerical stability. Regarding the size distribution, the sectional approach is widely used to simulate aerosol size distributions (Topping and Bane, 2022). However, limited bin resolution compromises the accuracy of size distribution and number concentration simulations (Kanakidou et al., 2005; Yu and Luo, 2009; Luo and Yu, 2011). Additionally, the hygroscopicity parameter (κ), derived from Köhler theory (Petters and Kreidenweis, 2007), is often simplified in chemical transport models, either by assuming a uniform

κ for organic aerosols in most global models (Fanourgakis et al., 2019) or various constant κ for different types of OA in regional models (Wang et al., 2019; Kuang et al., 2020). These simplifications introduce uncertainties in CCN simulations. However, in chemical transport models, it is difficult to assess the influence of these parameters on the uncertainties of CCN from SOA due to the mixing of SOA with other aerosol components, lack of direct observation constraints on all these parameters and complex factors influencing these parameters. Moreover, even without these simplifications, it is still not clear

how well we can model CCN concentration of SOA. Therefore, it is imperative to assess how well current understanding of SOA formation, i.e., chemical mechanisms can reproduce the SOA size distribution, chemical composition and κ. Such an assessment can only be achieved by comprehensive modelling of SOA formed in laboratory studies, where SOA precursors and formation conditions are well known and comprehensive measurements of SOA (mass, size distribution and CCN) are available.

Over the past two decades, based on comprehensive explicit gas-phase chemical mechanisms, numerous studies have modeled SOA formation by oxidation of biogenic VOCs (Jenkin, 2004; Xia et al., 2008; Capouet et al., 2008; Ceulemans et al., 2010; Chen et al., 2011; Valorso et al., 2011; Zuend and Seinfeld, 2012; Gatzsche et al., 2017; Galeazzo et al., 2021) or anthropogenic VOCs (Johnson et al., 2004, 2005; Hu et al., 2007; Camredon et al., 2007; Kelly et al., 2010; Xu, 2014; La et al., 2016; Lannuque et al., 2023) under different conditions in chamber environment. The MCM (Master Chemical Mechanism),

comprising 143 VOCs species and approximately 17,000 reactions, is the most widely used explicit mechanism (Jenkin et al., 1997, 2003; Saunders et al., 2003). Similar near-explicit or quasi-explicit chemical mechanisms include CACM (Caltech Atmosphere Chemistry Mechanism) (Griffin et al., 2002), GECKO-A (Generator for Explicit Chemistry and Kinetics of Organics in the Atmosphere) (Aumont et al., 2005; Camredon et al., 2007), PRAM (Peroxy Radical Autoxidation Mechanism) for the production of gas-phase Highly Oxygenated organic Molecules (HOMs) (Roldin et al., 2019), and other diversified

VOCs oxidation mechanisms constructed by different studies (Peeters et al., 2001; Capouet et al., 2004, 2008; Hu et al., 2007; Ceulemans et al., 2010). HOMs, a group of VOCs oxidation products formed through rapid autoxidation, play a critical role in SOA formation due to their low volatility and high oxygen content (Ehn et al., 2014; Bianchi et al., 2019). Modeling studies have demonstrated the importance of HOMs in SOA production from α-pinene and $\triangle^3$-carene ozonolysis reactions (Roldin et al., 2019; Xu, 2021; Luo et al., 2024; Thomsen et al., 2024). In the models simulating SOA formation, gas-particle partitioning

has been modeled based on thermodynamic absorption partitioning theory (Pankow, 1994) or dynamic gas-particle mass transfer following Raoult's law (Seinfeld and Pandis, 2016). Besides gas-phase reaction and gas-particle partitioning, particle-phase reactions, such as oligomerization and polymerization, have also been shown to significantly affect SOA speciation (Jenkin, 2004; Johnson et al., 2004, 2005; Xu, 2014; Hu et al., 2007; Chen et al., 2011; Galeazzo et al., 2021; Jia and Xu, 2021). For example, Hu et al. (2007) demonstrated that up to 70% of SOA mass from toluene photooxidation originated from

oligomers and polymers, underscoring the importance of particle-phase chemistry. Additionally, model studies have also indicated the importance of non-ideality and vapor pressure estimation methods in SOA simulations (Ceulemans et al., 2010; Kelly et al., 2010; Valorso et al., 2011; Zuend and Seinfeld, 2012).

     Despite advancements in chemical mechanisms and gas-particle partitioning models, SOA simulations still have limits. Most SOA modeling studies have focused on mass concentration and SOA yield, while far fewer studies have simulated the



chemical composition, particle size distribution and CCN concentrations (Jenkin, 2004; Johnson et al., 2004, 2005; Hu et al., 2007; Xia et al., 2008; Capouet et al., 2008; Chen et al., 2011; Xu, 2014). In general, the simulation of SOA chemical composition is relatively poor. For example, the average oxygen-to-carbon ratio (O:C) and hydrogen-to-carbon ratio (H:C) of simulated SOA showed significant gaps compared with measured SOA in different VOCs oxidation reactions (Chen et al., 2011). Moreover, few studies have simulated the particle size, which is crucial to SOA physical properties. Jia and Xu (2021)

developed the CSVA (Core-Shell box model for Viscosity dependent SOA) model, which simulates SOA size distribution evolution using a core-shell structure to account for viscosity effects in the gas-particle mass transfer. O'Meara et al. (2021) developed a python box model PyCHAM (CHemistry with Aerosol Microphysics in Python) for simulating aerosol chambers, which can also simulate particle size distribution. By using PyCHAM, Xu (2021) modeled SOA size distribution from α-pinene ozonolysis and showed a good envelope between the simulated and measured particle size evolution, though in the

simulation it was shifted to a later position, meaning the delayed particle growth compared to measurement. Furthermore, up to now, there is still a lack of simulation for hygroscopicity of SOA and CCN number concentration based on explicit chemical mechanisms.

In this study we simulated mass concentration, chemical composition, size distribution and CCN concentrations of SOA formed in α-pinene ozonolysis, a bench-mark system in SOA studies, using PyCHAM. We integrated MCM and PRAM

mechanisms and consider gas-particle partitioning, gas-wall partitioning, nucleation, coagulation, and particle wall loss to investigate SOA formation and growth. Simulated SOA mass, number concentration, chemical composition, size distribution, κ, and CCN number concentration are compared with measurement. This analysis aims to identify key chemical processes influencing SOA mass and chemical composition, and highlights the importance of accurate simulation of κ and particle size distribution in CCN simulations.

**2 Methods**

**2.1 SAPHIR Chamber and Experiment**

The α-pinene ozonolysis experiment was conducted in the SAPHIR (Simulation of Atmospheric PHotochemistry In a large Reaction) chamber at Forschungszentrum Jülich, Germany. SAPHIR is a 270 $m^3$ double-walled cylindrical Teflon chamber with a surface-to-volume ratio of ~1 $m^2$ $m^{-3}$, as previously described (Rohrer et al., 2005; Zhao et al., 2015a, b). The chamber

utilizes natural sunlight for illumination and features a louvre system to switch between light and dark conditions. For this study, the experiment was performed in the dark with the louvres closed. Prior to the experiment, the chamber was flushed with high-purity synthetic air (purity > 99.9999% $O_2$ and $N_2$). The experiment was conducted at a relative humidity (RH) of 37–79% and a temperature range of 291.2–299.1 K (Fig. S1). A total of 20 ppbv of α-pinene was introduced into the chamber, followed by the addition of 50 ppbv $O_3$ after 30 minutes to initiate organic chemistry. The experiment lasted approximately

8.5 hours, and no seed aerosols were used.

Temperature and RH were monitored continuously throughout the experiment. A Scanning Mobility Particle Sizer (SMPS) coupled with a Condensation Particle Counter (CPC, TSI3785) measured SOA mass and number concentrations and size distributions. A Cloud Condensation Nuclei Counter (CCNc, Droplet Measurement Technique, USA) measured CCN number concentrations at four supersaturations (SS): 0.19%, 0.37%, 0.55%, and 0.73%. The SS calibration and κ parameter

calculations followed Zhang et al. (2023). An Aerosol Mass Spectrometer (AMS) provided SOA chemical composition data, including O:C and H:C elemental ratios. $O_3$ concentrations were measured using a UV photometer $O_3$ analyzer (ANSYCO, model O341M). OH, $HO_2$, and $RO_2$ radical concentrations were quantified using a laser-induced fluorescence system (LIF) (Fuchs et al., 2012). VOCs were characterized using a Proton Transfer Reaction Time-of-Flight Mass Spectrometer (PTR-ToF-MS, Ionicon Analytik, Austria). Gas-phase oxygenated products from α-pinene ozonolysis, including HOMs, were





analyzed using a Chemical Ionization Atmospheric Pressure Interface Time-of-Flight Mass Spectrometer (CIMS, Tofwerk
       AG/Aerodyne Research, Inc.) with nitrate ($NO_3^-$) as the reagent ion ($NO_3^-$-CIMS).

**2.2 PyCHAM Box Modeling**

The α-pinene ozonolysis experiment was simulated using the PyCHAM (CHemistry with Aerosol Microphysics in Python)
model (O'Meara et al., 2021). PyCHAM was developed with two precursor models as platforms: the Microphysical Aerosol

Numerical model Incorporating Chemistry (MANIC) for multiphase processes (Lowe et al., 2009) and PyBox for Python-
       based parsing and automatic generation of chemical reaction modules (Topping et al., 2018). PyCHAM is designed to simulate
       aerosol chamber experiments, enabling comparisons between simulations and observations to improve process understanding
       for atmospheric applications.

       PyCHAM solves coupled ordinary differential equations for gas-phase chemistry, gas-particle partitioning, and gas-wall

partitioning following Jacobson (2005). Gas–particle partitioning follows the formulation of Zaveri et al. (2008):

$$\frac{dc_{i,g}}{dt} = -\sum_{j=1}^{N} k_{i,j}\left(C_{i,g} - x_{i,j}p_i^0 K_{v,j}\gamma_{i,j}\right), \tag{1}$$

$$\frac{dc_{i,j}}{dt} = k_{i,j}\left(C_{i,g} - x_{i,j}p_i^0 K_{v,j}\gamma_i\right), \tag{2}$$

where component $i$ partitions into size bin $j$ from the gas phase $g$, with $N$ total size bins. Here, $x$ is the particle-phase mole
fraction, $p^0$ is the pure component liquid (sub-cooled if necessary) vapour pressure, $K_v$ is the Kelvin factor and $\gamma$ is the activity

coefficient. The first-order mass transfer coefficient $k_{i,j}$ for component $i$ to size bin $j$ incorporates the Fuchs-Sutugin transition
       regime correction (Fuchs and Sutugin, 1971) and can be adjusted based on $\gamma$ and accommodation coefficient of individual
       component.

       Gas–wall partitioning follows an analogous framework:

$$\frac{dc_{i,g}}{dt} = -k_w\left(C_{i,g} - \frac{c_{i,w}}{c_w}p_i^0\gamma_i\right), \tag{3}$$

$$\frac{dc_{i,w}}{dt} = k_w\left(C_{i,g} - \frac{c_{i,w}}{c_w}p_i^0\gamma_i\right), \tag{4}$$

where $p_i^0$ is the liquid (sub-cooled if necessary) saturation vapour pressure of component $i$ and $\gamma_i$ is its activity coefficient on
the wall. $k_w$ ($s^{-1}$) accounts for gas- and wall-phase diffusion, turbulence, accommodation coefficient, and chamber surface-
area-to-volume ratio. $k_w$ was set to $2.2\times10^{-3}$ $s^{-1}$ according to experimental measurement (Guo et al., 2022). Meanwhile, $C_w$ (g
$m^{-3}$) represents wall adsorption/absorption properties, including effects of RH, surface area, diffusivity, and porosity. We

conducted a sensitivity analysis of SOA mass concentration to a series of $C_w$ values (Fig. S2), and SOA mass concentration
       showed no apparent change when $C_w$ increased by one magnitude from $1\times10^{-7}$ to $1\times10^{-6}$ g $m^{-3}$. Therefore, $C_w$ was set to $1\times10^{-6}$ g $m^{-3}$.

       PyCHAM also simulates microphysical processes, including coagulation, nucleation, and particle wall loss, which
       influence particle number evolution. Using a semi-implicit equation, coagulation process accounts for Brownian diffusion,

convective Brownian diffusion enhancement, gravitational collection, turbulent inertial motion, turbulent shear, and Vander
       Waals collisions (Jacobson, 2005), without adjustable parameters.

       Nucleation is modeled using a tuned Gompertz function to fit measured particle number size distributions during the
       initial reaction phase, without inferring mechanistic details:

$$P_1(t) = nuc_{v1}\left(\exp\left(nuc_{v2}\left(\exp\left(-\frac{t}{nuc_{v3}}\right)\right)\right)\right), \tag{5}$$

where $P_1$ (no. $cm^{-3}$) is the number concentration of new particles after time $t$ that enter the smallest size bin, and $nuc_{vn}$ represents
       the user-defined parameters which allow the amplitude ($nuc_{v1}$), onset ($nuc_{v2}$) and duration ($nuc_{v3}$) of the curve to be adjusted.
       Note that Eq. (5) is independent of chemistry. In this study, particle number concentrations were firstly fitted to CPC
       measurements during the initial 0.57 hours by setting $nuc_{vn}$ ($nuc_{v1}$ = 22403, $nuc_{v2}$ = -17.66, $nuc_{v3}$ = 317.88). Particle size range





of formed SOA was set as 0–500 nm, and the radius of newly nucleated particles was set as 10.9 nm according to lower limit

of the size range of SMPS. PyCHAM employs a sectional approach, dividing particles into a number of size bins (set as 128 in this study) and simulating size changes using the moving-center or full-moving approaches (Jacobson, 2005); the latter was adopted in this study. In this way, the constrained and subsequently simulated number concentrations excluding coagulation agree well with CPC measurements ($R^2 = 0.89$; NMB = 0.29%; Fig. S3), nevertheless, the particle size distribution simulation deviates significantly from measurement (Fig. S4).

Therefore, to accurately simulate CCN which depends on both particle number and size distribution, nucleation scheme was not used and the particle number size distribution was instead constrained using the particle size distribution measured by SMPS during the initial 0.6 hours in the model assuming the species $C_{20}H_{30}O_{17}$ to represent low-volatile HOMs dimers as seed particle. The vapor pressure of $C_{20}H_{30}O_{17}$ at normal temperature calculated by default method of Nannoolal et al. (2008) is $2.14 \times 10^{-29}$ Pa, which is extremely low to act as a seed aerosol. The lower and upper boundaries and mean radii of size bins

were constrained according to SMPS measurements (9.6–437.2 nm) and size bin number was 106. Under this configuration, the simulated SOA mass concentration and chemical composition were significantly influenced by the presence of assumed seed particles. Therefore, to balance the simulation of chemical composition and particle size, the simulation of SOA mass concentration and chemical composition are based on the scheme of nucleation, as in Sect. 3.1. While as for particle number concentration and size distribution, the simulation adopts the scheme of constraining SMPS size distribution and number

concentration of seed particles together, as in Sect. 3.2. Subsequently, the κ is derived from the chemical composition of SOA displayed in Sect. 3.1. This was combined with the particle size distribution described in Sect. 3.2 to calculate CCN number concentrations, as in Sect. 3.3.

Particle wall deposition is simulated using either the McMurry and Rader (1985) model or a customized size-dependent deposition rate:

$D_\mathrm{p} < D_\mathrm{p,flec}$

$$\log_{10}\left(\beta\left(D_\mathrm{p}\right)\right) = \log_{10}\left(D_\mathrm{p,flec}\right) - \log_{10}\left(D_\mathrm{p}\right)\nabla_\mathrm{pre} + \beta_\mathrm{flec}, \tag{6}$$

$D_\mathrm{p} \geq D_\mathrm{p,flec}$

$$\log_{10}\left(\beta\left(D_\mathrm{p}\right)\right) = \log_{10}\left(D_\mathrm{p}\right) - \log_{10}\left(D_\mathrm{p,flec}\right)\nabla_\mathrm{pro} + \beta_\mathrm{flec}, \tag{7}$$

where $D_{p,flec}$ marks the inflection diameter for deposition rates, and $\beta_{flec}$ gives the deposition rate ($s^{-1}$) at this inflection point.

$\nabla_\mathrm{pre}$ and $\nabla_\mathrm{pro}$ represent the log-log slopes of deposition rate versus diameter before and after the inflection point. This study used a uniformed value ($\beta_{flec} = 2.37 \times 10^{-5}$ $s^{-1}$) based on the measured particle loss rates without considering the size dependence.

Moreover, other parameters such as time series of temperature (291.2–299.1 K) and RH (37–79%), and dilution rate ($9 \times 10^{-6}$ $s^{-1}$) during the experiment were constrained according to measurements.

The gas-phase chemical mechanism for α-pinene we used in PyCHAM draw upon previous studies as MCM coupled

with PRAM mechanisms (Roldin et al., 2019; O'Meara et al., 2021; Luo et al., 2024; Thomsen et al., 2024). PyCHAM currently lacks explicit treatment of particle-phase reactions and dissolution, which are hence not considered in our simulation. The simulated α-pinene concentrations agree with measurement ($R^2 = 0.99$) in this study (Fig. S5), indicating the capability to describe gas-phase chemistry of α-pinene ozonolysis by PyCHAM with MCM + PRAM mechanism.

### 2.3 Hygroscopicity Parameter (κ) and CCN Concentration

The hygroscopicity parameter (κ) of bulk SOA was calculated using the UManSysProp module (Topping et al., 2016), an open-source tool for predicting molecular and atmospheric aerosol properties (https://github.com/loftytopping/UManSysProp_public/). UManSysProp estimates pure component vapor pressures, critical properties, sub-cooled densities of organic molecules; activity coefficients for mixed inorganic-organic liquid systems; hygroscopic growth factors and CCN activation potential of mixed inorganic–organic aerosol particles with associated κ–



Köhler values (Kreidenweis et al., 2005); and absorptive partitioning calculations with/without a treatment of non-ideality. Users input molecular information as SMILES (Simplified Molecular Input Line Entry System) strings, and UManSysProp automatically extracts relevant information for calculations.

In PyCHAM, UManSysProp predicts molecular weight, pure liquid density, and liquid saturation vapor pressure for individual components. Default methods include Girolami (1994) for liquid density and Nannoolal et al. (2008) for vapor 
pressure estimation. Since PyCHAM does not currently include κ prediction, we further calculated the κ values for bulk SOA under ideal condition given molar concentrations, vapor pressures, densities, temperatures, dry particle sizes, and a surface tension of 72 mN m$^{-1}$. The critical activated dry particle size ($D_{p,dry}$) for CCN activation at different SS levels was derived from the κ–Köhler equation (Petters and Kreidenweis, 2007):

$$\frac{RH}{100} = \frac{D_{p,wet}^3 - D_{p,dry}^3}{D_{p,wet}^3 - (1-\kappa)D_{p,dry}^3} \exp\left(\frac{4\sigma_s M_w}{RT\rho_w D_{p,wet}}\right), \quad (8)$$

where $\sigma_s$ is the surface tension of the wet particle at the solution-air interface, $M_w$ is the molecular weight of water, $R$ is the ideal gas constant, $T$ is temperature, $\rho_w$ is the density of water, and $D_{p,wet}$ is the diameter of the wet particle.

Particles larger than $D_{p,dry}$ can act as CCN and $D_{p,dry}$ decreases with increasing SS. Then CCN number concentrations at different SS ($N_{CCN,SS}$) were calculated by integrating the simulated particle number size distribution $PNSD(D)$ over size bins exceeding $D_{p,dry}$:

$$N_{CCN,SS} = \int_{D_{p,dry}}^{D_{max}} PNSD(D)dD, \quad (9)$$

where $D$ is dry particle diameter, and $D_{max}$ is the maximum $D$.

### 3 Results and Discussion

### 3.1 Simulation of Particle Formation and Chemical Composition

Figure 1 shows the simulated and measured SOA mass concentrations. The simulated SOA mass concentration exhibited 
a high correlation ($R^2 = 0.97$) with measurements and the two showed similar temporal trends, characterized by a rapid increase within the first ~3 hours followed by a gradual decline. The simulated decline rate closely matched measurements (Fig. 1b), except for a faster decrease around ~6 hours. This discrepancy is likely attribute to a rapid temperature increase during this period (Fig. S1), as elevated temperature can cause organic compounds to evaporate from the particle phase, reducing SOA formation (Donahue et al., 2006; Xia et al., 2008; Ceulemans et al., 2010). Simulated particle-phase HOMs accounted for ~43% 
of total SOA mass concentration (Fig. 1a), highlighting the important contribution of HOMs to SOA and necessity of including HOMs formation in the chemical mechanisms as done here by coupling MCM with the PRAM mechanism. Without PRAM mechanism, the onset of SOA growth was significantly delayed and mass concentration was obviously lower. The significant contribution of HOMs to SOA is consistent with previous studies. For example, Roldin et al. (2019) found ~50% of SOA mass from α-pinene ozonolysis with ammonium sulfate seeds originated from HOMs condensation. Gatzsche et al. (2017) reported 
HOMs contributed up to 65% of SOA mass during early-stage α-pinene ozonolysis and accounted for about 27% of the total SOA mass throughout the simulation.

Despite good correlation with measured SOA concentration, simulated SOA mass concentration was consistently underestimated (19.1% ± 10.4%). The underestimation can be attributed to incomplete description of gas-phase chemistry, gas-particle partitioning, and/or particle-phase chemistry. The gas-phase chemistry such as α-pinene losses (Fig. S5) and 
HOMs composition are well generally simulated (Fig. S6), showing the bimodal distribution of monomers (m/z 230–380) and dimers (m/z 400–550). This also indicates the capability of PRAM mechanism to effectively characterize gas-phase HOMs formation, as shown by Roldin et al. (2019). The gas-particle partitioning mass transfer coefficient in the model also influences SOA mass concentration. PyCHAM allows adjustment of activity and accommodation coefficients to modify this parameter. However, the simulated SOA mass concentration rise rate closely matched measurements (Fig. 1b), indicating that the mass




transfer coefficient adequately represents gas-particle partitioning. Therefore, the underestimation is likely attributed to missing particle-phase chemistry. Particle-phase accretion reactions or oligomerization processes can produce larger, higher-molecular-weight species, increasing SOA mass concentration (Pun and Seigneur, 2007; Kroll and Seinfeld, 2008; Hallquist et al., 2009). The influence of particle-phase reaction on SOA mass concentration has been reported in previous studies. For example, Hu et al. (2007) attributed up to 70% of SOA mass to oligomers and polymers in toluene photooxidation, highlighting

the importance of particle-phase reactions. By adjusting branching ratio for HOMs formation and gas-wall partitioning parameters, Xu (2021) performed a good consistency between simulated and measured SOA mass concentrations from α-pinene ozonolysis. However, the simulated SOA mass concentration was still underestimated from their data even after considering OH-initiated secondary autoxidation, which might be due to the lack of particle-phase reactions in their simulation. Particle-phase reactions, including heterogeneous reactions on aerosol surfaces and in particles of organic compounds, can

also alter SOA properties, such as solubility, viscosity, hygroscopicity, and optical properties (Farmer et al., 2015; Shrivastava et al., 2017; Jia and Xu, 2021). For example, Galeazzo et al. (2021) found that missing autoxidation and particle-phase reactions in α-pinene ozonolysis simulations led to underestimated SOA viscosity.

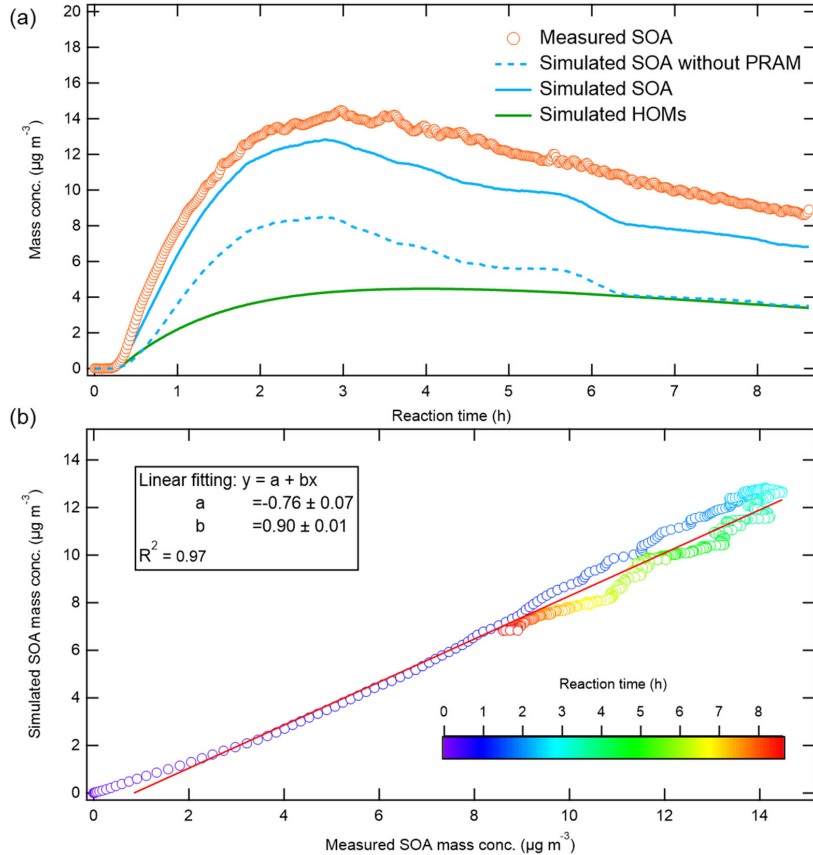

**Figure 1: (a) Mass concentrations (µg m⁻³) of simulated (blue line) and measured (circles) SOA, alongside simulated particle-phase**
**HOMs mass concentration (green line). The blue dashed line refers to simulated SOA without considering PRAM mechanism. (b) Scatter plot of measured versus simulated SOA mass concentrations, with a linear fit (red line). The coefficients a and b represent the intercept and slope, respectively, and R² denotes the correlation coefficient. Colors indicate the reaction time of the experiment.**

Simulated O:C and H:C ratios of SOA were compared with measurements as AMS can only provide bulk O:C and H:C ratios of SOA rather than molecular chemical composition (Fig. 2). Compared to measured O:C (0.44 ± 0.03) and H:C (1.35

± 0.02) ratios, the simulated average O:C (0.58 ± 0.03) and H:C (1.64 ± 0.00) ratios were overestimated. Without PRAM



mechanism, the simulation showed lower O:C and higher H:C as a result of less gas-phase HOMs formation, while still overestimating the two ratios. Overestimation of H:C (21.2% ± 2.1%) and O:C (32.4% ± 2.2%) ratios is likely attributed to the absence of particle-phase reactions in our simulations as mentioned above, emphasizing the importance of particle-phase chemistry in determining SOA chemical composition. The difference between modelled and measured O:C and H:C has also
been reported by previous studies. Using similar gas-phase chemical mechanism, Roldin et al. (2019) reported similar overestimations in modelled H:C ratios, while modelled O:C ratios agree with measurements in their studies. Chen et al. (2011) observed overestimated O:C and H:C ratios in α-pinene ozonolysis simulations. However, HOMs are not included in the mechanism of their study. They proposed a chemical mechanism involving particle-phase decomposition of organic hydroperoxides and subsequent oligomerization involving free radicals to explain the discrepancies. Although HOMs
formation was included in our study, the lack of similar particle-phase reactions can still contribute the overestimation of H:C and O:C ratios.

Regarding the temporal changes, both the measured and simulated average O:C increased gradually over time before stabilizing, which is possibly attributed to the dilution of SOA concentration and/or the increasing fraction of particle-phase HOMs during the experiment (Fig. S7). In contrast, the measured H:C decreased over time, while the simulated H:C showed
no apparent variations. Particle-phase reactions, such as oligomerization, typically generate high-molecular-weight compound and alter O:C and H:C ratios of organic matter (Kroll and Seinfeld, 2008; Hallquist et al., 2009), and their absence in simulations likely contributed to the discrepancy between modelled and measured time series of O:C and H:C ratios.

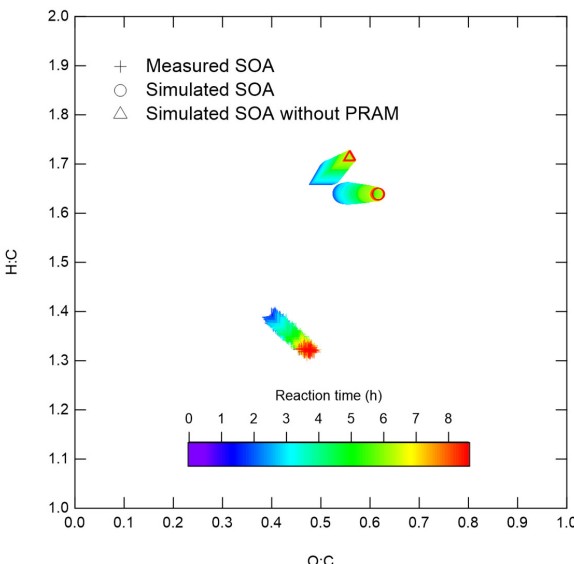

**Figure 2: Chemical composition represented by the average O:C and H:C ratios of simulated (circles) and measured (crosses) SOA,**
**as well as simulated SOA without considering PRAM mechanism (triangles). The symbol sizes refer to SOA mass concentrations.**
**Valid SOA measurement data were available from 0.45 hours due to the low SOA concentrations before that.**

### 3.2 Simulation of Particle Number Concentrations and Size Distribution

Despite a slight underestimation (7.3% ± 2.8%) of the simulated number concentration since the model run freely following the constrained particle number size distribution (an intercept of -1657 # cm⁻³), the simulated particle number concentration
showed good agreement with measurements (Fig. 3), with a linear fit yielding a slope of 1.04 and a correlation coefficient of $R^2 = 0.99$. Our result is similar to the report of Xu (2021) which also exhibited an underestimation of particle number concentration when coagulation was included.



Particle number concentration is primarily influenced by microphysical processes such as nucleation, coagulation, particle wall loss, and gas-particle partitioning. In our simulation for particle number, the initial phase of particle growth was constrained by particle size distribution of SMPS measurement instead of setting nucleation parameters, as detailed in Sect. 2.2. Besides, particle loss rate to wall was fully constrained by measurements, and gas-particle partitioning showed good performance in simulated SOA mass concentration. To explore the impact of coagulation on particle number concentration and size distribution, we tested including and excluding coagulation in our simulation (Fig. S8-10). Coagulation, particularly among nanoparticles, increases collision probabilities, leading to the formation of larger particles and reduction in particle number concentration, and a shift in size distribution toward larger diameters (Jacobson, 2005; Seinfeld and Pandis, 2016). In this study, the simulated number concentration without coagulation declined more slowly than measurements throughout the whole simulation period (Fig. S8), resulting in an overestimation of 14.4% ± 7.9%, which is much more deviated from measured than that when including coagulation. In spite of using a semi-implicit coagulation equation (Jacobson, 2005) without adjustable parameters, our result suggests that the rates of coagulation of particles are well represented in PyCHAM model.

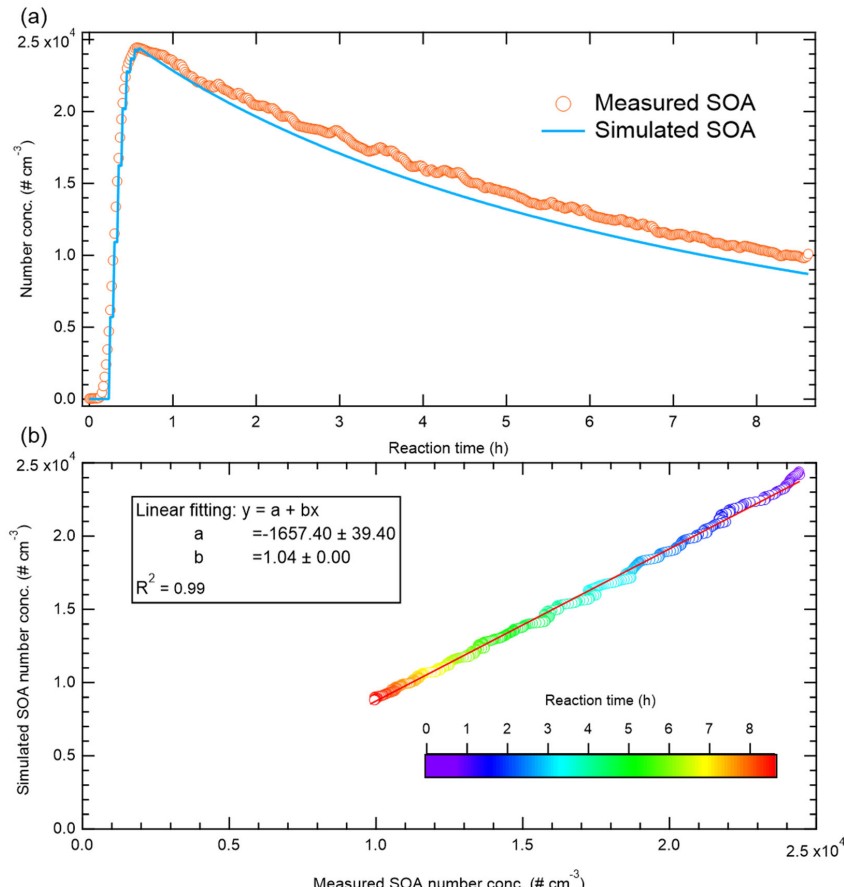

**Figure 3: (a-b) Same as Fig. 1, but for SOA number concentration (# cm⁻³). Scatters from the initial 0.6 hours were excluded, as the number concentration during this period was fitted to SMPS measurements and assumed to match measurements perfectly. Note that the coagulation is included.**

To further evaluate simulation accuracy of particle size, the geometric mean diameter of simulated SOA was calculated (Fig. 4), which showed good agreement with measurements ($R^2 = 0.96$) and a slight underestimation of 1.9% ± 2.8%, indicating an excellent reproduction of the central position of size distribution. Without PRAM mechanism, the geometric mean diameter of SOA was lower due to the absent production of larger molecules. When coagulation was excluded, the geometric mean



diameter was underestimated by 8.8% ± 1.1% (Fig. S9), and was merely underestimated by 6.9% ± 2.9% compared to the result including coagulation, suggesting that particle growth during the reaction was primarily driven by condensation, overweighing the influence of coagulation on the size distribution. As a result, the size distribution showed only minor changes due to coagulation, which is in contrast with the large changes in particle number concentrations (Fig. S8).

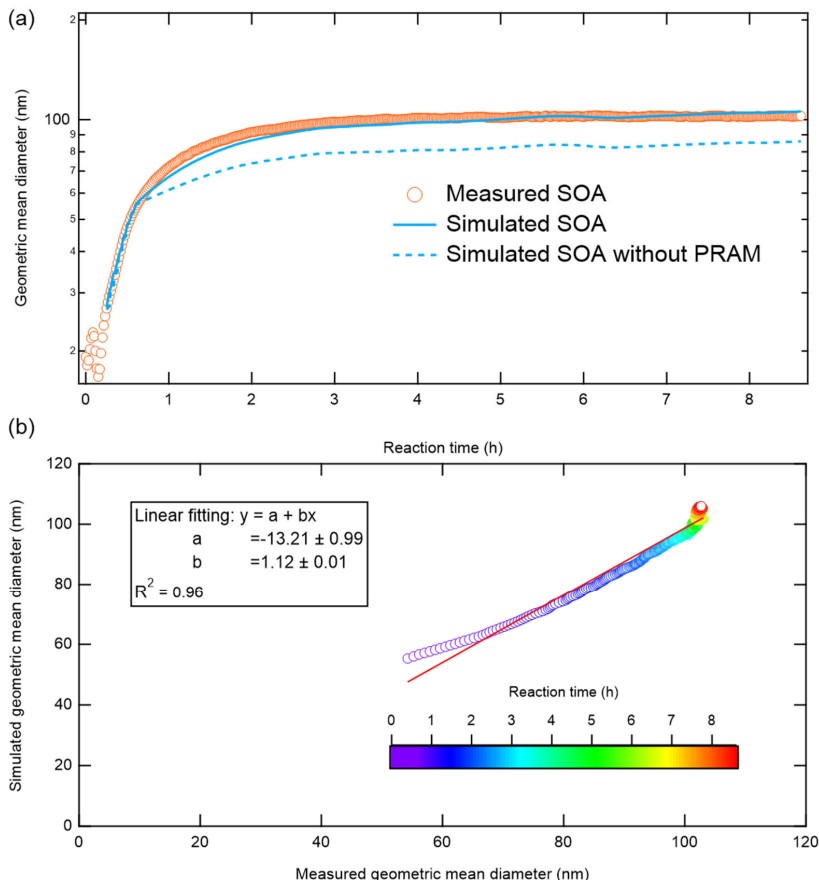

**Figure 4: (a-b) Same as Fig. 1, but for geometric mean diameter (nm) of SOA. And blue dashed line refers to simulated SOA without considering PRAM mechanism. Note that the coagulation is included, and scatters from the initial 0.6 hours were excluded as particle size distribution during this period was fitted to SMPS measurements.**

Figure 5 compares the measured and simulated number size distributions of SOA, with shaded areas representing simulations and contour lines denoting measurements. The simulation effectively captured the trend of particle growth, and the simulated size range generally aligned with measurements. However, simulated particle size distribution showed flatter and wider distribution patterns than measurement, even though we constrained them by SMPS during the initial 0.6 hours. When nucleation parameters were assigned to simulate initial particle growth in PyCHAM, instead of constraining the initial particle size distribution with SMPS measurements, the simulated size distribution exhibited greater deviation from measured data (Fig. S4). Xu (2021) assigned the nucleation parameters for the initial particle growth in PyCHAM, and simulated smaller particle sizes and a slower particle growth than measurements. These results indicate that no matter whether nucleation or seed particles are used to specify the initial number particle size distribution in PyCHAM, the model currently still needs improvement to better represent the evolution of particle size. Nevertheless, as the model has generally well reproduced the particle size and number concentrations, it can be used for subsequent CCN simulations. When coagulation was excluded in our study (Fig. S10), the simulated particle size distribution showed some odd spikes in the early stage and more deviation





from the measurement especially in the subsequent growth stage, demonstrating the reliable representation of coagulation in PyCHAM model.

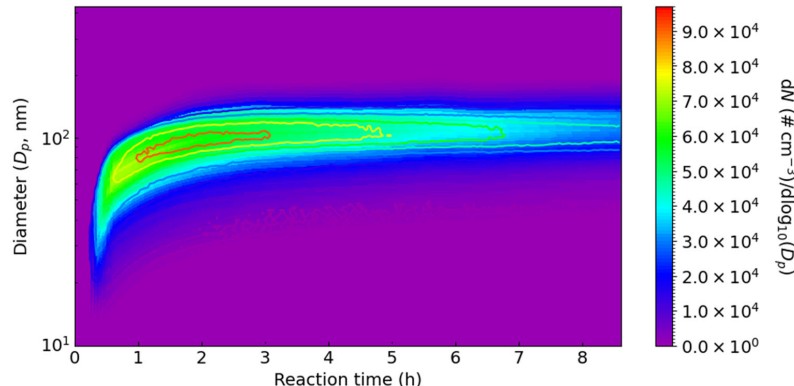


**Figure 5: The time evolution of number size distributions (dN/dlog₁₀Dₚ) of measured (contour lines) and simulated (shaded areas) SOA. Note that the coagulation is included.**

### 3.3 Simulation of κ and CCN Concentrations

We calculated the hygroscopicity parameter (κ) of bulk SOA under ideal condition using the UManSysProp and compared

with measurement (Fig. 6). The measured κ generally increased with higher SS, which is possibly attributed to the size dependence of chemical composition, consistent with previous monoterpene oxidation studies (Zhao et al., 2015a; Zhang et al., 2023). In contrast, the simulated κ was independent of SS and did not show a significant size dependence. From 50.6 to 84.8 nm of particle size, the κ derived from the chemical composition corresponding to size only decreased by 0.002, indicating weak dependence of simulated chemical composition on particle size, which is different from the measurement (Table S1).

The simulated κ increased during the first hour before stabilizing at 0.172 ± 0.003. Compared to measurements, the simulated κ was underestimated by 20.7% ± 4.9% at higher SS levels (0.73% and 0.55%), overestimated by 18.6% ± 5.9% at SS = 0.19%, and showed the closest agreement at SS = 0.37%, with an overestimation of 9.6% ± 8.5%. Overall, the simulated κ agreed well with measurements, particularly at lower SS levels.

Field observation and laboratory studies have shown that κ of SOA decreases with increasing molecular weight under

supersaturated conditions (Kuwata et al., 2013; Wang et al., 2019). We computed the average molecular weight of SOA (Fig. S11) and found that the molecular weight decreased rapidly during the initial phase before stabilizing, explaining the measured and simulated increase in κ in the early stage in the experiment and its subsequent leveling off.



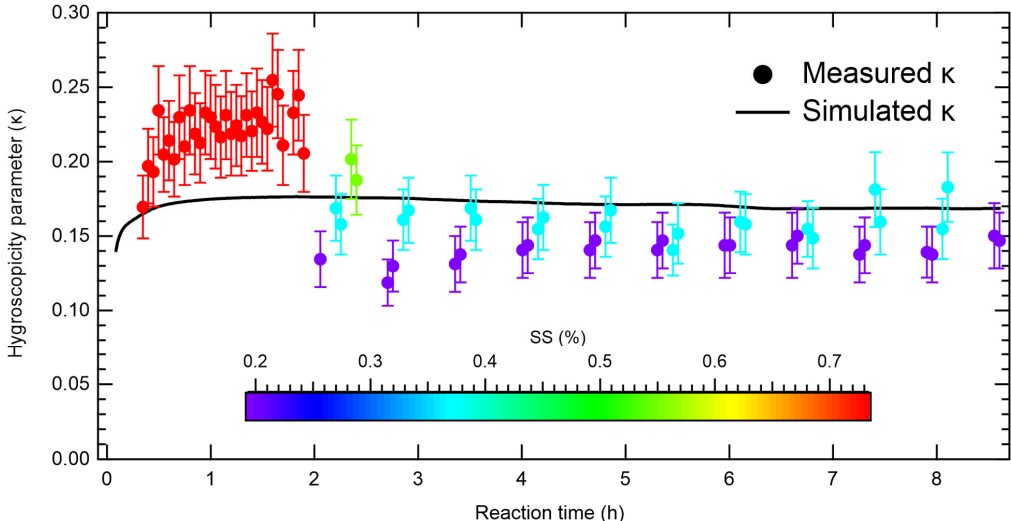

**Figure 6: The hygroscopicity parameter (κ) of simulated (line) and measured (solid circles) SOA with standard deviation (error bar)**
**at different supersaturation (SS).**

CCN number concentrations at different SS levels were derived using κ and particle size distribution (Fig. 7). At higher
SS levels (0.73% and 0.55%), the simulated CCN number concentration closely matched measurements throughout the
reaction time ($R^2$ = 0.88–0.99), with the exception of a more rapid increase during the initial period at SS = 0.73%. Although
the simulated κ were significantly underestimated compared to the corresponding measured κ at these two SS levels, leading
to an overestimation of the CCN critical activated dry particle size ($D_{p,dry}$), the close agreement between the simulated and
measured geometric mean diameter of SOA (Fig. 4) resulted in a compensatory effect. Specifically, the wider and flatter size
distribution patterns in the simulation compensated the impact of the lower κ, coincidentally yielding a good agreement in
CCN number concentrations at these two SS. At SS = 0.37%, the simulated CCN number concentration showed close
agreement with the measured CCN ($R^2$ = 0.98), with a mean bias of -3.9% ± 1.9%. This consistency in CCN concentration
corresponded to the smallest discrepancy in κ at this SS. Although the simulated κ was slightly higher, a concurrently slightly
lower simulated particle size resulted in a comparable CCN value. As for SS = 0.19%, the simulated CCN number
concentration exhibited the largest discrepancy among all SS levels, consistently overestimating the measured values by a
factor of >4 throughout the reaction. This overestimation was primarily attributed to the excessively wider and flatter simulated
particle size distribution and a pronounced overestimation of κ at this lowest SS which led to an underestimation of $D_{p,dry}$.

To further investigate the influence of κ values and particle size distribution on CCN simulation results, we systematically
examined different scenarios by:

(1) maintaining the κ, and adopting the number particle size distribution of SMPS measurement or the 8-bin particle size
scheme simulation which is generally used in conventional 3D models (such as WRF-Chem), or

(2) maintaining the number particle size distribution, and applying the κ derived by measurements or a fixed κ of 0.1
generally used in 3D models.

Through separate modifications of κ values and particle size distribution, we calculated the corresponding CCN number
concentrations. As shown in Fig. S12, when the particle size distribution was varied while maintaining a constant κ, the 106-
bin particle size scheme employed in this study demonstrated superior performance to the conventional 8-bin approach across
various SS levels. The 106-bin results showed closer agreement with CCN number concentrations derived from SMPS-
measured size distributions, particularly during the initial growth phase of CCN activation. At higher SS levels (0.73% and
0.55%), however, CCN number concentrations exhibited less sensitivity to the bin numbers option, indicating negligible



dependence on the number size distribution, which is partly due to the $D_{p,dry}$ became sufficiently low at higher SS that most particles can act as CCN.

When κ was varied while maintaining the number size distribution (Fig. S13), CCN number concentrations calculated

385   using κ from UManSysProp showed excellent agreement with those derived from observationally inferred κ across all SS levels, except for an overestimation at the lowest SS. In contrast, the fixed κ = 0.1 scheme systematically underestimated CCN number concentrations, with the discrepancy increasing at lower SS. These findings indicate that the simulation of CCN concentration acted by SOA relies on the accurate representations of κ and particle size distribution, particularly for lower SS levels.

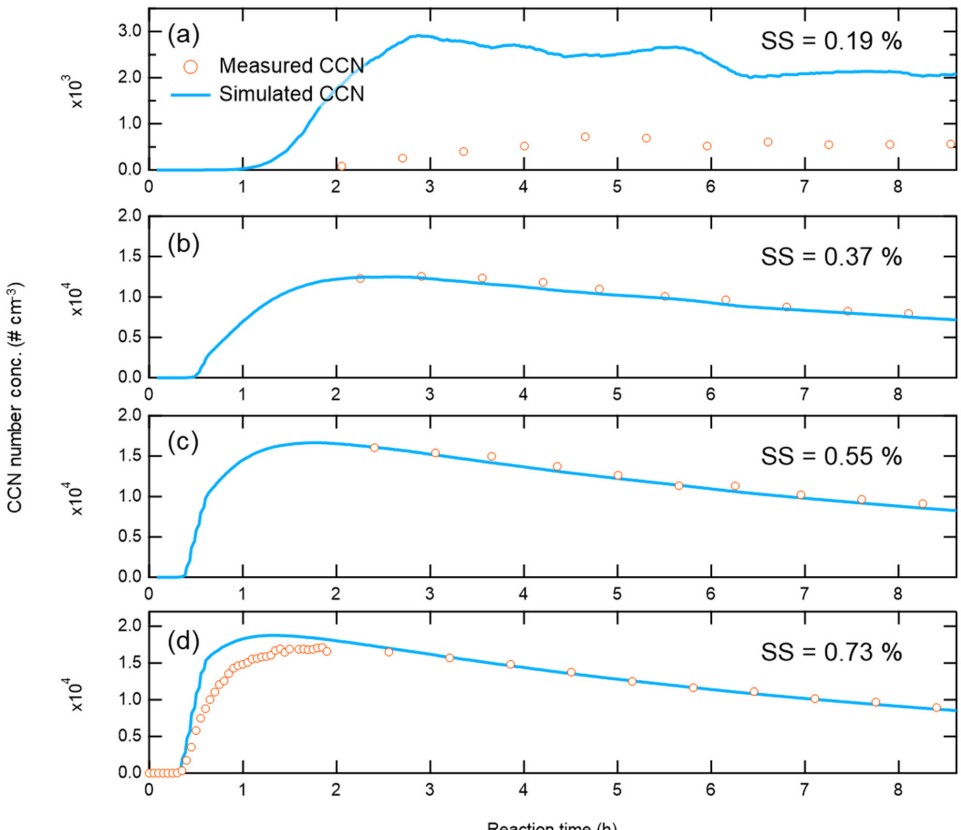

**Figure 7: (a-d) The measured (circles) and simulated (lines) CCN number concentrations (# cm$^{-3}$) at different SS.**

## 4 Conclusions and implications

In this study, we simulated mass concentration, number concentration, chemical composition (O:C and H:C ratios) and size distribution of SOA from α-pinene ozonolysis by coupling MCM and PRAM explicit chemical mechanisms in a box

395   model PyCHAM. We further simulated CCN number concentrations at a series of SS using hygroscopicity parameter (κ) of SOA calculated by UManSysProp according to κ-Köhler theory, and particle size distribution.   Compared to chamber experimental measurement, the SOA mass concentration was well reproduced with an underestimation of 19.1% ± 10.4%. O:C and H:C ratios were overestimated by 32.4% ± 2.2% and 21.2% ± 2.1%, respectively. Besides, the time evolution of elemental ratios was inconsistent with measurement. These discrepancies were likely attributed to the missing particle-phase

400   reactions during the simulation, as gas-phase chemistry including α-pinene decay and HOMs composition were generally well



represented. Moreover, the contribution of simulated particle-phase HOMs to SOA mass concentration was significant (~43%), underscoring the critical role of HOMs in SOA production.

By constraining SMPS particle size distribution and number concentration during the initial reaction time, the simulated SOA number concentration exhibited a good agreement with measurement ($R^2 = 0.99$). The simulated particle size distribution, however, showed wider and flatter patterns, suggesting the necessity to better represent size evolution in the PyCHAM model in the future. Moreover, the simulated κ showed overestimation (18.6% ± 5.9%) at the lowest SS (0.19%) and underestimation (20.7% ± 4.9%) at higher SS levels (0.73% & 0.55%), with the closest agreement at SS = 0.37%. Correspondingly, the simulated CCN number concentrations had varying levels of bias across different SS levels. Notably, at higher SS levels (0.73% and 0.55%), the underestimated κ values led to an overprediction of $D_{p,dry}$, yet this bias was compensated by the wider and flatter particle size distribution in the simulations. As a result, the CCN number concentrations at these two SS levels exhibited excellent consistency with measurements ($R^2 = 0.88$–99). At SS = 0.37%, the slightly overestimated κ was balanced by the slightly lower simulated particle size, yielding closely matched CCN values ($R^2 = 0.98$). In contrast, at SS = 0.19%, the substantial overestimation of κ and the resulted underestimated $D_{p,dry}$, coupled with the excessively wider and flatter particle size distribution, collectively led to a significant overestimation of CCN number concentrations.

To further quantify the individual contributions of κ and size distribution on CCN, we conducted comparison analyses by using different κ schemes and different number of particle size bins to represent the number size distribution. It is found that accurate representation of both κ and particle size distribution is critical for reliable CCN simulations acted by SOA, particularly at lower SS levels (<0.4%). At higher SS levels (>0.4%), however, the sensitivity of CCN predictions to these parameters decreases as SS increases, as the smaller $D_{p,dry}$ at higher SS render most particles to act as CCN regardless of variations in κ or size distribution.

This study advances previous research by simulating CCN number concentrations from SOA using explicit chemical mechanisms (MCM + PRAM) for the first time to our knowledge. We comprehensively examined how SOA mass concentration, chemical composition, hygroscopicity, and size distribution collectively influence CCN formation. Although our simulation, like earlier studies, exhibits biases chemical composition, we highlight the importance of missing particle-phase processes in SOA production. Crucially, the simulated size distribution and hygroscopicity impact the CCN predictions especially at lower supersaturations. This study highlights that accurately representing SOA hygroscopicity and size distribution is key to reducing modelled CCN uncertainties.

**Data availability**

All the data in the figures of this study are available upon request to the corresponding author (dfzhao@fudan.edu.cn).

**Supplement**

**Author contributions**

DZ conceptualized the study. ZS performed the model simulation. CZ analyzed the measured CCN data and calculated the hygroscopicity parameter. HM analyzed the mass spectra data of gas-phase HOMs. ZS wrote the manuscript. ZS and DZ edited the manuscript with the input from all co-authors. All the co-authors discussed the results and commented on the manuscript.

**Competing interests**

The authors declare that they have no conflict of interest.



**Acknowledgements**

Zhen Song, Chenqi Zhang, Hongru Shen, Hao Ma, and Defeng Zhao would like to thank the funding support of Shanghai Pilot Program for Basic Research-Fudan University 21TQ1400100 (22TQ010) and the National Natural Science Foundation of China (No. 42575109). We thank Thomas F. Mentel for the support of this study. We gratefully acknowledge Simon Patrick O'Meara for updating the code and function of PyCHAM and the support of the xml file in simulation.

**Financial support**

This research has been supported by Shanghai Pilot Program for Basic Research-Fudan University 21TQ1400100 (22TQ010) and the National Natural Science Foundation of China (No. 42575109).

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
