# Peer review of "Explicit simulation of chemical composition, size distribution and cloud condensation nuclei of secondary organic aerosol from $\alpha$ -pinene ozonolysis"

_EGUsphere, 2025_

## Author Comment (AC1)

We thank the anonymous reviewer for the comments on our manuscript. The comments and suggestions are greatly appreciated. All the comments have been addressed and we believe that the revisions based on these comments have improved the quality of our manuscript. Below please find our responses to the comments one by one and the corresponding revisions made to the manuscript. The original comments are in italics. The revised parts of the manuscript are in blue here and can be followed in the revised manuscript with track changes with line numbers indicated.

Responses to reviewer 1:

*Reviewer: 1*

*Comments:*

*Secondary organic aerosol (SOA) may contribute significantly to cloud condensation nuclei (CCN), yet relevant research of explicit simulation remains relatively limited. While existing SOA modeling studies predominantly concentrate on mass concentration, this work specifically investigates the CCN activity of SOA, thereby advancing our understanding of SOA's role in CCN formation. My specific comments are as follows:*

*(1) Although the paper is titled " Explicit simulation of chemical composition, size distribution and cloud condensation nuclei of secondary organic aerosol from α-pinene ozonolysis", it only provides detailed descriptions for the size distribution and CCN simulations, with inadequate description on the chemical composition of SOA. For instance, the number of species involved in gas-particle partitioning in the model remains unspecified. Furthermore, no information is provided regarding whether the gaseous concentrations of these species were characterized with experimental observations or have undergone laboratory validation. The authors should provide a list of substances involved in gas-particle transformation in supplement file.*

**Response:**

Accepted. In our experiments SOA chemical composition was measured by AMS (e.g. O:C and H:C ratios). Although PyCHAM can simulate the gas–particle partitioning of all species involved in the chemical mechanism (MCM+PRAM) and calculate their concentrations across particle-phase size bins, we lack direct molecular-level information on individual compounds in the particle-phase. For this reason, the main text focuses on discussing the elemental composition (ratios of different elements) of SOA.

Gas-phase species participating in gas-particle partitioning, mainly oxygenated organic molecules (OOMs), were measured using a nitrate-CIMS.

In the revised manuscript, we have clearly described the measurements of organic compounds participating in gas-particle partitioning as follows (line 139-141).

"Gas-phase oxygenated organic molecules (OOMs) participating in gas-particle partitioning, including HOMs, were measured using a chemical ionization atmospheric pressure interface time-of-flight mass spectrometer (CIMS, Tofwerk AG/Aerodyne Research, Inc.) with nitrate ($NO_3^-$) as the reagent ion ($NO_3^-$-CIMS)."

We have also included in the Supplement the chemical formulas of all organic species (including HOMs) produced by MCM and PRAM mechanisms (Table S1), and the corresponding text were added in the main manuscript in line 234-235.

"The detailed chemical species formulas produced by MCM and PRAM mechanisms are shown in Table S1."

Table S1 is shown below:

| Chemical species in MCM or PRAM | | | |
|---|---|---|---|
| CH3O2 | C9PAN2 | C721PAN | C18H26O4 |
| CH3O | C85CO3H | C721CO3H | C88O2 |
| CH3NO3 | C85OOH | NORPINIC | C718CO3 |
| HCHO | C86OOH | C721OOH | C87O2 |
| CH3O2NO2 | C511OOH | C722OOH | NC826O2 |
| CH3OOH | C7PAN3 | C44OOH | C18H27O6NO3 |
| CH3OH | C235C6CO3H | C811NO3 | C20H30O8 |
| CO23C4CHO | CO235C6OOH | C516O | C20H31O6NO3 |
| BIACETO2 | APINBO | C516OOH | C19H28O9 |
| CH3CO3 | APINBNO3 | C10H15O2O2 | C19H29O6NO3 |
| HCOCH2CO3 | APINBOOH | LIMOOA | C18H26O9 |
| CO23C4CO3 | APINBCO | LIMALAO2 | C18H27O7NO3 |
| C5PAN9 | NAPINAO2 | LIMALBO2 | C20H30O9 |
| CO23C4CO3H | NAPINBO2 | C10H17O3O2 | C20H31O7NO3 |
| CO23C3CHO | NAPINAO | BPINENE | C19H28O10 |
| HCOCH2CHO | NAPINAOOH | LIMONENE | C19H29O7NO3 |
| HCOCH2O2 | NAPINBO | CARENE | C18H26O10 |
| C3PAN2 | NC101CO | C10H15O4O2 | C18H27O8NO3 |
| HCOCH2CO3H | NAPINBOOH | C10H15O3O2 | C20H30O10 |
| HCOCH2CO2H | NC101O2 | C10H15O5O2 | C20H31O8NO3 |
| GLYOX | NC101O | C10H15O6O2 | C19H28O11 |
| BIACETO | NC102O2 | C10H15O7O2 | C19H29O8NO3 |
| BIACETOOH | NC102O | C10H15O8O2 | C18H26O11 |
| BIACETOH | NC71O2 | C10H15O9O2 | C18H27O9NO3 |
| HOCH2CO3 | NC71O | C10H15O10O2 | C20H30O11 |
| HOCH2CHO | NC71CO | C10H15O11O2 | C20H31O9NO3 |
| PHAN | NC101OOH | C10H15O12O2 | C19H28O12 |
| HOCH2CO3H | NC102OOH | C10H15O2O | C19H29O9NO3 |
| HOCH2CO2H | NC71OOH | C10H15O3O | C18H26O12 |
| ACETOL | NC72O2 | C10H15O4O | C18H27O10NO3 |
| MGLYOX | NC72O | C10H15O5O | C20H30O12 |
| CH3COCH2O2 | NC61CO3 | C10H15O6O | C20H31O10NO3 |
| CH3COCH2O | NC72OOH | C10H15O7O | C19H28O13 |
| HCOCO | NC6PAN1 | C10H15O8O | C19H29O10NO3 |

| | | | |
|---|---|---|---|
| HCOCO3 | NC61CO3H | C10H15O9O | C18H26O13 |
| HCOCO3H | APINCO | C10H15O10O | C18H27O11NO3 |
| HCOCO2H | APINCNO3 | C10H15O11O | C20H30O13 |
| HMVKAO2 | C720O2 | C10H15O12O | C20H31O11NO3 |
| HMVKAO | APINCOOH | C10H14O3 | C19H28O14 |
| HMVKANO3 | APINCOH | C10H14O4 | C19H29O11NO3 |
| HMVKAOOH | HCC7CO | C10H14O5 | C18H26O14 |
| HO12CO3C4 | C720O | C10H14O6 | C18H27O12NO3 |
| CO2H3CHO | C720NO3 | C10H14O7 | C20H30O14 |
| CO2H3CO3 | C720OOH | C10H14O8 | C20H31O12NO3 |
| C4PAN6 | C720OH | C10H14O9 | C19H28O15 |
| CO2H3CO3H | C719O2 | C10H14O10 | C19H29O12NO3 |
| PAN | C719O | C10H14O11 | C18H26O15 |
| CH3CO3H | C719NO3 | C10H14O12 | C18H27O13NO3 |
| CH3CO2H | C719OOH | C10H14O13 | C20H30O15 |
| HCOCH2O | C719OH | C10H15O2NO3 | C20H31O13NO3 |
| HCOCH2OOH | APINOOA | C10H15O3NO3 | C19H28O16 |
| CH3COCH3 | APINOOB | C10H15O4NO3 | C19H29O13NO3 |
| HYPERACET | C107O2 | C10H15O5NO3 | C18H26O16 |
| CHOC3COCO3 | C109O2 | C10H15O6NO3 | C18H27O14NO3 |
| CHOC3COO2 | C107O | C10H15O7NO3 | C20H30O16 |
| CHOC3COO | C108O2 | C10H15O8NO3 | C20H31O14NO3 |
| CHOC3COPAN | C108O | C10H15O9NO3 | C19H28O17 |
| CHOC3COOOH | C108NO3 | C10H15O10NO3 | C19H29O14NO3 |
| C413COOOH | C717O2 | C10H15O11NO3 | C18H26O17 |
| C4CODIAL | C717O | C10H15O12NO3 | C18H27O15NO3 |
| C312COCO3 | C717NO3 | C10H16O4iso1 | C20H30O17 |
| CHOCOCH2O2 | C107OOH | C10H16O5iso1 | C20H31O15NO3 |
| CHOCOCH2O | C107OH | C10H16O6iso1 | C19H28O18 |
| C312COPAN | C108OOH | C10H16O7iso1 | C19H29O15NO3 |
| C312COCO3H | C108OH | C10H16O8iso1 | C18H26O18 |
| ALCOCH2OOH | C717OOH | C10H16O9iso1 | C18H27O16NO3 |
| C33CO | C717OH | C10H16O10 | C10H16O3 |
| H1CO23CHO | C109O | C10H16O11 | C10H17O5O2 |
| APINENE | C89CO3 | C10H16O12 | C10H17O4O2 |
| APINAO2 | C920CO3 | C10H16O13 | C10H17O6O2 |
| APINBO2 | C109OOH | C10H16O14 | C10H17O7O2 |

| | | | |
|---|---|---|---|
| APINCO2 | C109OH | C20H30O5 | C10H17O8O2 |
| APINAO | C109CO | C20H30O6 | C10H17O3O |
| APINANO3 | C920O2 | C20H30O7 | C10H17O4O |
| PINAL | C920O | C923CO3 | C10H17O5O |
| APINAOOH | C921O2 | LIMAO2 | C10H17O6O |
| APINBOH | C921O | LIMCO2 | C10H17O7O |
| C96O2 | C922O2 | LIMALO2 | C10H16O4iso2 |
| C96CO3 | C922O | LIMBO2 | C10H16O5iso2 |
| PINALO2 | C621O2 | C20H31O4NO3 | C10H16O6iso2 |
| C96O | C621O | BPINAO2 | C10H16O7iso2 |
| C96NO3 | H1C23C4CHO | BPINBO2 | C10H16O8iso2 |
| C97O2 | H1C23C4O2 | BPINCO2 | C10H16O9iso2 |
| C97O | H1C23C4CO3 | C918CO3 | C10H17O3NO3 |
| C98O2 | H1C23C4O | C20H31O5NO3 | C10H17O4NO3 |
| C98O | H1C23C4PAN | NLIMO2 | C10H17O5NO3 |
| C98NO3 | HC23C4CO3H | NLIMALO2 | C10H17O6NO3 |
| C614O2 | H1C23C4OOH | NC91CO3 | C10H17O7NO3 |
| C614O | C920PAN | NBPINAO2 | C10H17O8NO3 |
| C614NO3 | C920CO3H | NBPINBO2 | C10H18O5 |
| PINALO | HOPINONIC | C19H28O5 | C10H18O6 |
| PINALNO3 | C920OOH | C19H28O6 | C10H18O7 |
| C106O2 | C921OOH | C19H28O7 | C10H18O8 |
| C106O | C922OOH | C19H28O8 | C10H18O9 |
| C106NO3 | C621OOH | C923O2 | C10H18O10 |
| C716O2 | APINBOO | C924O2 | C20H34O6 |
| C716O | C89CO2 | C816CO3 | C20H34O7 |
| CO13C4CHO | C89O2 | NORLIMO2 | C20H34O8 |
| C10PAN2 | C89O | LMKAO2 | C20H35O5NO3 |
| PERPINONIC | C89NO3 | LMKBO2 | C20H35O6NO3 |
| PINONIC | C810O2 | C926O2 | C19H32O6 |
| C96OOH | C810O | C817CO3 | C19H32O7 |
| C96OH | C810NO3 | LMLKAO2 | C19H32O8 |
| NORPINAL | C514O2 | LMLKBO2 | C19H32O9 |
| C97OOH | C514O | C823CO3 | C19H33O6NO3 |
| C97OH | C514NO3 | C925O2 | C18H30O6 |
| C98OOH | C89PAN | NOPINAO2 | C18H30O7 |
| C98OH | C89CO3H | NOPINBO2 | C18H30O8 |

| | | | |
|---|---|---|---|
| C614OOH | C89CO2H | NOPINCO2 | C18H30O9 |
| C614OH | C89OOH | NOPINDO2 | C18H30O5 |
| C614CO | C89OH | C918O2 | C18H31O7NO3 |
| PINALOOH | C810OOH | C9DCO2 | C20H34O9 |
| PINALOH | C810OH | C915O2 | C20H35O7NO3 |
| C106OOH | C514OOH | C917O2 | C19H32O10 |
| C106OH | C514OH | C919O2 | C19H33O7NO3 |
| C716OOH | C811CO3 | C914O2 | C18H30O10 |
| C716OH | C811O2 | C916O2 | C18H31O8NO3 |
| CO235C6CHO | C811O | C88CO3 | C20H34O10 |
| H3C25C6O2 | C812O2 | C87CO3 | C20H35O8NO3 |
| H3C25C6CO3 | C812O | C822CO3 | C19H32O11 |
| H3C25C6O | C813O2 | NLMKAO2 | C19H33O8NO3 |
| H3C2C4CO3 | C813O | C19H29O5NO3 | C18H30O11 |
| H3C2C4PAN | C813NO3 | C18H26O5 | C18H31O9NO3 |
| H3C2C4CO3H | C516O2 | C18H26O6 | C20H34O11 |
| H3C2C4CO2H | C811CO3H | C18H26O7 | C20H35O9NO3 |
| H3C25C6PAN | PINIC | C729CO3 | C19H32O12 |
| H3C25C5CHO | C811PAN | C816O2 | C19H33O9NO3 |
| H3C25CCO3H | C811OOH | C817O2 | C18H30O12 |
| H3C25CCO2H | C811OH | C826O2 | C18H31O10NO3 |
| H3C25C6OOH | C721CHO | C822O2 | C20H34O12 |
| H3C25C6OH | C812OOH | C818O2 | C20H35O10NO3 |
| C85O2 | C812OH | C823O2 | C19H32O13 |
| C85CO3 | C813OOH | C819O2 | C19H33O10NO3 |
| C85O | C813OH | C727CO3 | C18H30O13 |
| C86O2 | CO13C3CO2H | C731CO3 | C18H31O11NO3 |
| C86O | C721O2 | C824O2 | C20H34O13 |
| C511O2 | C721CO3 | C820O2 | C20H35O11NO3 |
| C511O | C721O | C18H26O8 | C19H32O14 |
| CO235C5CHO | C722O2 | C825O2 | C19H33O11NO3 |
| CO235C6CO3 | C722O | C821O2 | C18H30O14 |
| CO235C6O2 | C44O2 | C732CO3 | C18H31O12NO3 |
| CO235C6O | C44O | C8BCO2 | C10H18O4 |

*(2) Accurate simulation of CCN critically depends on both number concentration and particle size distribution. Notably, the authors employed two distinct methods for number concentration: when modeling CCN, they utilized*

*observation-derived fitting results, whereas for SOA mass simulation, they adopted a nucleation scheme based on C20H30O17 molecule. Why were these two methods applied separately? Are the simulation results from these two approaches comparable?*

**Response:**

When modeling SOA mass concentration and chemical composition, we used nucleation scheme. In the current version of PyCHAM, the nucleation process can only be constrained using three parameters that determine the initial growth of particle number concentration. However, the particle size distribution (PSD) during the early nucleation stage cannot be set in nucleation scheme. As a result, the simulated PSD exhibits a clear bias in peak position relative to the observations. Figure R1 illustrates the PSD at reaction time of 2 h. Because the accuracy of CCN number concentration depends on both the SOA size distribution and the hygroscopicity parameter ($\kappa$), any bias in the PSD directly affects the CCN simulation. To improve the representation of early growth, we constrained the initial PSD using the SMPS measurements and assuming a seed aerosol, i.e. using seed scheme. The PSD at reaction time hour 2 is shown below (Fig. R2); this approach performs better than the nucleation scheme in simulating PSD. However, since this approach requires specifying an explicit seed species, we selected $C_{20}H_{30}O_{17}$—an organic molecule with sufficiently low vapor pressure—as the seed. In this case, the simulated SOA mass and O:C ratio are thus influenced by the assumed seed composition, ultimately increasing the discrepancy with observations (Fig. R3). Consequently, the overall performance on chemical composition is worse than the nucleation scheme.

[Figure]

**Fig. R1: The measured and nucleation scheme-simulated particle size distribution (PSD) at the reaction time of 2 h.**

[Figure]

**Fig. R2: Same as Fig. R1, with the addition of the PSD simulated by the seed scheme.**

[Figure]

**Fig. R3: O:C and H:C distributions of SOA measured experimentally and simulated using the two schemes (nucleation and seed).**

To obtain accurate CCN predictions, bulk κ of SOA was calculated from the chemical composition derived using the nucleation scheme, and was subsequently combined with the PSD from the seed scheme to compute CCN. Admittedly, this hybrid approach may lack coherence and general applicability. To assess how each scheme influences CCN results, we first applied the nucleation scheme consistently for both SOA κ and CCN simulations (Fig. R4), and compared the resulting CCN with those presented in the main text (Fig. 7). This isolates the influence of PSD on CCN. The results indicate that the PSD of both schemes obtain similar CCN number concentrations, which are close to observations at supersaturation (SS) = 0.55% and 0.73%. Under the nucleation scheme, CCN at SS = 0.37% is slightly overestimated, and CCN at SS = 0.19% is initially higher than observations but gradually decreases toward zero.

Next, we applied the seed scheme consistently for both SOA κ (Fig. R5) and CCN simulations (Fig. R6) and compared these CCN values with those in the original manuscript. This isolates the influence of κ on CCN. Both schemes demonstrate similar CCN prediction performance across four SS, though CCN simulated by the seed scheme was slightly

lower than that of the combined scheme as a result of lower simulated κ (Fig. R5).

[Figure]

Fig. R4: CCN number concentrations (# cm⁻³) measured experimentally and simulated using κ and PSD from nucleation scheme.

[Figure]

**Fig. R5: Measured and simulated (using the seed scheme) SOA κ.**

[Figure]

**Fig. R6: Same as Fig. R4, but for CCN simulated by κ and PSD from seed scheme.**

Overall, if the nucleation scheme is applied alone, the simulated PSD performs worse than that obtained with the combined approach, resulting in larger bias of CCN concentrations at the two lower SS. In contrast, applying the seed

scheme alone leads to worse simulations of initial SOA mass concentration, chemical composition, and κ due to the assumed composition of seed species. However, because the PSD remains relatively accurate, the resulting CCN concentrations are similar to those from the combined approach. Therefore, in this study we adopted the combined approach, which reconciles the simulations of both chemical composition and PSD while minimizing bias in CCN predictions.

We have added the description about the influence of two independent schemes on CCN predictions in Sect. 3.4 (line 452-465).

"**3.4 Discussion of the influence of individual schemes (nucleation vs. seed) on CCN predictions**

To demonstrate the rationale for the combined approach - using κ from the nucleation scheme together with PSD from the seed scheme - a detailed analysis of the effect of applying each scheme independently on the CCN simulations is implemented.

As shown in Fig. S21, CCN calculated using the κ by the nucleation scheme (Fig. 6) and PSD by the same scheme (Fig. S4) at SS = 0.55% and 0.73% were comparable to those from the combined-scheme approach. However, at SS = 0.37%, CCN was moderately overestimated, and at SS = 0.19% the predicted CCN was initially higher than the measurements and then decreased toward zero. In contrast, CCN calculated using the κ from the seed scheme (Fig. S22) combined with its PSD (Fig. 5) produced lower CCN across all four SS (Fig. S23), leading to a worse performance than that of the combined-scheme approach.

Overall, if the nucleation scheme was applied alone, the simulated PSD performed worse than that obtained with the combined approach, resulting in deviations of CCN concentrations at the two lower SS. In contrast, applying the seed scheme alone led to worse simulations of initial SOA mass concentration, chemical composition, and κ due to the assumed composition of seed species. However, because the PSD remained relatively accurate, the resulting CCN concentrations were similar to those from the combined approach."

Figure S21-23 in the Supplement correspond to Fig. R4-6 here.

*(3) Line 116: Please specify the exact model of the DMA in the SMPS. Also, provide the specific model of the AMS, and similarly, specify the models of other equipment used.*

**Response:**

Accepted. We have now specified the instrument models for the DMA, AMS, and CCN measurements in the revised manuscript as follows (line 122-124).

"A scanning mobility particle sizer (SMPS, TSI DMA3081/TSI CPC3785) measured SOA mass and number concentrations and size distributions over the range 9.82–429.4 nm. A cloud condensation nuclei counter (CCN100, Droplet Measurement Technique, USA) measured CCN…"

And line 134-136:

"A high-resolution time-of-flight aerosol mass spectrometer (HR-ToF-AMS, Aerodyne Research Inc., DeCarlo et al., 2006) provided SOA chemical composition data, including O:C and H:C elemental ratios."

*(4) Line 119: The authors state that "The SS calibration and κ parameter calculations followed Zhang et al. (2023)," but later in the results section, it is mentioned that κ was measured. The authors should explain how κ was measured in*

*the experimental section.*

**Response:**

We apologize for the ambiguity. The measured κ are determined using Scanning Mobility CCN Analysis (SMCA) method (Moore et al., 2010). The detailed procedure is referred to our previous studies (Zhao et al., 2015, 2016). Briefly, for each of the four SS, CCN number concentration and total particle number concentration (CN) in each SMPS size bin are measured in parallel by coupling a DMA with a CCN counter and CPC. Particles pass through the DMA and the outgoing air is split into two paths connecting to the CCN counter and CPC. For each particle size, the CN and CCN concentrations are used to calculate the activation fraction (CCN/CN). Then, CCN/CN is fitted with Gaussian error function and the critical activation dry diameter ($D_{crit}$) at the set SS is the turning point of this function. Then κ parameter at four SS is derived from κ–Köhler equation given different SS and corresponding $D_{crit}$ (Petters and Kreidenweis, 2007). These κ values are what we refer to as "measured κ" in the main text.

In contrast, the simulated κ values are calculated directly from the modeled SOA molecular concentrations, vapor pressures, density, dry diameter, temperature, and surface tension (Kreidenweis et al., 2005). Therefore, the simulated κ does not depend on SS, unlike the observation-derived κ values.

We have added a detailed description of the derivation of the measured κ values in Section 2.1 as follows to clarify this point (line 125-134).

"Based on parallel measurements of CCN and total particle number (cloud nuclei; CN) for each size bin in a continuous flow, the critical activation particle size ($D_{crit}$) at each SS was determined using the Scanning Mobility CCN Analysis (SMCA) method (Moore et al., 2010; Zhao et al., 2015a, 2016). Briefly, CN and CCN concentrations for each size bin were used to calculate the CCN activation fraction (CCN/CN). Before computing CCN/CN, the measured CCN and CN concentrations were corrected for multiple charged particles. Then, CCN/CN for each charge class was then fitted using a Gaussian error function, and the turning point of this function was taken as $D_{crit}$ at the specific SS. For each SS, at least three full scans were performed, and the resulting $D_{crit}$ were averaged. The SS calibration followed Zhao et al. (2016) and Zhang et al. (2023). Then κ parameter at four SS was derived from κ–Köhler equation given different SS and corresponding $D_{crit}$ (Petters and Kreidenweis, 2007). The error bars for κ were estimated from the standard deviation of $D_{crit}$ across three duplicate scans."

*(5) Line 140: Please provide the specific formula used to calculate $k_{i,j}$, as well as the range and basis for the values of γ and α in this study.*

**Response:**

Accepted. We have added the explicit expression for the mass-transfer coefficient $k_{i,j}$ in the main text (Zaveri et al., 2008). Because no well-established data of activity coefficient γ were available for our experimental conditions, we only simulated the idealized conditions. Non-ideality was neglected, and all activity coefficient γ were set to 1. In our simulations, the accommodation coefficients α for all species were assumed to be 1. These parameter choices have now been clearly stated in the revised manuscript as follows (line 158-166).

"mass accommodation coefficient ($\alpha_i$) of individual component:

$$k_{i,j} = 4\pi \overline{R_{p,j}} D_{g,i} N_j f(Kn_{i,j}, \alpha_i), \quad (3)$$

where $\overline{R_{p,j}}$ (cm) is mean wet radius of particles in bin $j$; $D_{g,i}$ (cm$^2$ s$^{-1}$) is gas diffusivity of species $i$; $N_j$ (cm$^{-3}$) is

the number concentration of particles in bin $j$; $\alpha_i$ means the chance that component $i$ can stick to a particle surface when collision happens. In our simulation, $\alpha_i$ for all components were set to 1. And $f(Kn_{i,j}, \alpha_i)$ is the transition regime correction factor to the Maxwellian flux as a function of the Knudsen Number:

$$f\left(Kn_{i,j}, \alpha_i\right) = \frac{0.75\alpha_i\left(1 + Kn_{i,j}\right)}{Kn_{i,j}\left(1 + Kn_{i,j}\right) + 0.283\alpha_i Kn_{i,j} + 0.75\alpha_i}, (4)$$

$$Kn_{i,j} = \frac{\lambda_i}{R_{p,j}}, \quad (5)$$

where $\lambda_i$ is the mean free path."

And line 155-156:

"Because no well-established data of $\gamma$ were available for our experimental conditions, we only simulated the idealized conditions (i.e. $\gamma$ for all components were set to 1)."

*(6) Line 165: The authors mention that the aerosol particle size was divided into 128 bins, but later state that it was divided into 106 bins. This inconsistency should be clarified, and the aerosol bin division should be explained in detail in the methods section.*

**Response:**

Accepted. We apologize for the ambiguity. We used 128 size bins in the nucleation scheme, following the sensitivity analysis of O'Meara et al. (2021), who recommends using 128 bins when accurate representation of the PSD is important. In this configuration, the particle size range is set to 1.8–500 nm, with an initial logarithmic bin width of 0.019 nm. Although the upper bound is 500 nm, the simulated dN/dlogD$_p$ is distributed within 9.2–146.2 nm (with values beyond this range being zero).

In the seed-based scheme we used the 106 size bins because the size distribution was constrained by SMPS measurement, which has 106 size bins. The size range is 9.82–429.4 nm, and the average initial logarithmic bin width is approximately 0.016 nm, similar to the bin width in the nucleation scheme. The simulated dN/dlogD$_p$ is mostly distributed within 12.0–215.6 nm (accounting for 99.9% of the total), which is close to that of the nucleation scheme (9.2–146.2 nm).

We have clarified the rationale for choosing 128 or 106 size bins in the revised manuscript as follows (line 196-198).

"As recommended by O'Meara et al. (2021) that a more detailed 128 size bins should be adopted when the number PSD is important, we set the bin number to 128 and employed the full-moving approach to simulate size evolution."

And line 206-208:

"The lower and upper boundaries and mean radii of each size bin and bin number were set according to SMPS (9.82-429.4 nm size range and 106 size bins)."

*(7) Lines 196-198: The authors compare measured and simulated values of α-pinene to indicate the capability of PyCHAM with the MCM + PRAM mechanism to describe the gas-phase chemistry of α-pinene ozonolysis. To validate the model's performance in simulating the MCM gas-phase reactions after incorporating the HOMs module, comparing only the reactants is insufficient. It is recommended to also compare the temporal evolution of other major product concentrations, particularly the simulation performance for HOMs.*

**Response:**

Accepted. In the revised version, we further compared the temporal evolution of gas-phase HOMs during the initial 10 min of reaction, including monomers ($C_{10}H_{15}O_8$, $C_{10}H_{14}O_{11}$, $C_{10}H_{16}O_{11}$) and dimers ($C_{20}H_{30}O_{10}$, $C_{20}H_{30}O_{12}$, $C_{20}H_{30}O_{15}$). The model generally well simulated the temporal trend of HOMs, although there are some biases in the absolute concentrations (Fig. R7). Together, these results indicate that the gas-phase chemistry of α-pinene ozonolysis in this study is reasonable.

[Figure]

**Fig. R7: Measured and simulated time evolution of gas-phase HOMs mixing ratio (ppb) during the initial 10 min of reaction.**

We have added the discussion on the gas-phase products HOMs in the main text (line 232-234).

"and the temporal trends of gas-phase products HOMs (Fig. S8) are well captured, though there are some biases in the absolute concentrations, indicating the capability to describe gas-phase chemistry of α-pinene ozonolysis by PyCHAM with MCM and PRAM mechanisms."

Figure S8 in the main text corresponds to Fig. R7 here.

*(8) The authors attribute the overestimation of simulated O/C and H/C ratios to the lack of consideration of particle-phase reactions in the model. However, in Figure S6, the simulated HOMs are generally higher than the measured values, especially for ions with m/z above 400. Yet, the total SOA mass concentration is simulated well, implying that the simulation underestimates other components while overestimating HOMs. Clearly, the overestimation of HOMs would lead to higher O/C ratios. Additionally, the authors should analyze the reasons for the overestimation of HOMs in the simulation compared to observations (Figure S6).*

**Response:**

In the original manuscript, we compared the simulated and observed gas-phase HOMs only after normalizing both spectra to their respective maximum signal. We have compared the simulated and observed volume-concentration mass spectra of gas-phase HOMs (Fig. R8). Although the simulated total concentration of gas-phase HOMs during the first 5 min of the experiment (0.011 ppb) is slightly underestimated compared to the measurement (0.014 ppb), the results indicate that the simulation reproduces the observed HOMs species (m/z) reasonably well. Specifically, the concentration levels of dimers are captured closely, while those of monomers are underestimated, particularly at m/z < 300. Our spectra pattern is similar to the findings of Roldin et al. (2019), especially for dimers, who also showed a slight underestimation of monomers. Furthermore, the fractions of HOMs monomers and dimers are also well captured (Fig. R9). These findings together with Fig. R7 suggest that the gas-phase chemical mechanism employed in the model is generally reasonable.

While SOA mass concentration exhibited similar temporal trends and high correlation coefficient, it was underestimated by 19.1%. Given the reasonable performance of gas-phase chemistry and gas-particle partitioning, we attributed the discrepancies in SOA mass concentration and O:C and H:C possibly to the absence of particle-phase chemistry in the model. And the slight underestimation of gas-phase HOMs would not lead to higher O:C.

[Figure]

**Fig. R8: Measured and simulated gas-phase HOMs mass spectra averaged over the first 5 min of experiment, during which gas-phase HOMs were rapidly accumulated and particle-phase concentrations were low.**

[Figure]

**Fig. R9: Pie charts of (a) measured and (b) simulated gas-phase HOMs monomer and dimer fractions averaged over the first 5 min of the reaction.**

In the revised manuscript, we have revised the discussion as follows (line 276-280).

"The gas-phase chemistry, including the loss of α-pinene (Fig. S7) and the composition of HOMs, is generally well reproduced (Fig. S9). The model reproduces the bimodal distributions of HOM monomers (m/z 230-380) and dimers (m/z 400-550), although the concentration of monomers is underestimated, especially below m/z 300. It also reasonably captures the fractions of HOM monomers and dimers (Fig. S10), while showing a slight underestimation of dimers in the simulation."

Figure S9-10 in the main text correspond to Fig. R8-9 here.

*(9) It is difficult to observe the differences between the simulated and observed particle size distributions in Figure 5. It is recommended to supplement the figure with a two-dimensional curve showing the particle number concentration as a function of particle size at a specific time.*

**Response:**

Accepted. The geometric mean diameter of SOA reflects only the general tendency of the size distribution and does not capture information about peak width. To address this limitation, we have added two-dimensional $dN/dlogD_p$ plots for reaction hours 2, 4, 6, and 8 (Fig. R10 below and Fig. S14 in the revised manuscript). These plots provide a more clear comparison and illustrate the differences between the simulated and observed size distributions (i.e., the simulated distributions are flatter and broader).

We have added the following text in the revised manuscript (line 377-379).

"Figure S15 presents the $dN/dlog_{10}D_p$ versus PSD at 2, 4, 6, and 8 h of reaction time, clearly illustrating that the simulated PSDs were broader and flatter than measurement."

The Fig. S15 in the main text corresponds to Fig. R10 here.

[Figure]

**Fig. R10: (a-d) Number particle size distribution (dN/dlogDp) at reaction hours 2, 4, 6, and 8.**

*(10) How were the κ values in Figure 6 measured? This is not explained in the text. Furthermore, why does the measured κ value show a sudden decrease at the second hour, while the simulated value does not exhibit such a change? As shown in the figure, κ values differ under different SS conditions, so what SS was used to determine the simulated κ?*

**Response:**

Accepted. As mentioned in the response to comment 4, we have added a detailed description of how κ is measured.

The sharp decrease in the measured κ around hour 2 occurs because the SS switched from 0.73% to 0.19%, leading to a much lower CCN number concentration and consequently affecting the inferred critical activation dry diameter ($D_{crit}$) and the resulting κ. After hour 2, κ values are showed only for SS = 0.19% and 0.37%, because the $D_{crit}$ derived from fitting CCN/CN activation curves at SS = 0.73% and 0.55% have too large uncertainties as almost all particles are activated.

Since the measured κ values are derived directly from CCN number concentrations, they necessarily correspond to specific SS. The dependence of κ on SS may result from the dependence of chemical composition on particle size as the $D_{crit}$ at different SS are different as we discussed in the manuscript (Zhao et al., 2015; Zhang et al., 2023). In contrast, the simulated κ values of bulk SOA are computed from the modeled SOA molecular composition, vapor pressure, density, dry diameter, temperature, and surface tension (Kreidenweis et al., 2005). As shown in Table S2, simulated chemical composition and κ of SOA did not show dependences on particle size in the size range of the $D_{crit}$ measured at various SS. Therefore, simulated κ did not correspond to a specific SS.

In the revised manuscript, we have added the following text to clarify this problem (line 398-399).

"The sudden decrease in κ measured at ~2 h of reaction is attributed to the decrease of the set SS from 0.73% to 0.19%. In contrast, the simulated κ was formula-based and did not correspond to specific SS."

And line 410-411:

"After ~2 h, κ values were showed only for SS = 0.19% and 0.37%, because the $D_{crit}$ derived from fitting CCN/CN activation fraction curves at SS = 0.73% and 0.55% had too large uncertainties as almost all particles were activated."

*(11) When SS = 0.19%, the simulated CCN concentration is much higher than the measured value. The authors attribute this overestimation to the wider and flatter particle size distribution in the simulation. Why does this overly broad particle size distribution not cause significant deviations under other high SS conditions?*

**Response:**

The procedure for calculating simulated CCN is as follows. Using the κ-Köhler equation, we first compute the critical activation dry diameter ($D_{crit}$) corresponding to each SS based on the simulated κ values. We then integrate the particle number size distribution above $D_{crit}$ to obtain the CCN number concentration for each SS. Thus, the simulated CCN depends directly on both $D_{crit}$ and the PSD.

As shown in the figure R11 below (Fig. S17 in the revised manuscript), we present the simulated and observed $D_{crit}$ values for the four SS levels, along with their corresponding size distributions, before and after hour 2 of the experiment. For SS = 0.19% and 0.37%, the simulated κ values are overestimated, leading to underestimated $D_{crit}$ (located to the left of the observed $D_{crit}$; panels (a) and (b)). Conversely, for SS = 0.55% and 0.73%, the simulated κ values are underestimated, yielding overestimated $D_{crit}$ (to the right of the observations; panels (c) and (d)).

At SS = 0.19%, the combination of underestimated $D_{crit}$ and the simulated size distribution being broader and flatter in peak height leads to a substantial overestimation of CCN. In contrast, for the other SS levels, despite the broader simulated size distributions, the simulated and observed $D_{crit}$ values are very similar and lie to the left of the dN/dlogDp peak. As a result, the flatter and broader simulated size distributions tend to offset the effect of the $D_{crit}$ differences, producing CCN number concentrations that deviate only slightly from the observations.

In the revised manuscript, we have revised the following text to clarify this problem (line 413-431).

"Figure S18 presents the PSD and $D_{crit}$ at four SS levels corresponding to time points before and after 2 h, providing additional context for interpreting the discrepancies between simulated and measured CCN. At the higher SS levels of 0.73% and 0.55%, the simulated CCN number concentrations closely matched the measurements throughout the reaction ($R^2$ = 0.88-0.99), except for a more rapid increase during the initial period at SS = 0.73%. Although κ was underestimated at these SS, leading to slightly overestimated $D_{crit}$, the simulated and measured $D_{crit}$ were still very similar and both positioned to the left of the PSD peak (Fig. S18c and d). Under these conditions, the broader and flatter simulated PSD introduced a compensating effect, resulting in simulated CCN concentrations that were very close to the measurements. The slight overestimation of CCN before 0.6 h at SS = 0.73% was primarily attributable to the low bias in simulated κ, since the simulated and measured PSD were identical during this period.

At SS = 0.37%, the simulated CCN number concentrations also agreed closely with measured CCN ($R^2$ = 0.98) with a mean bias of -3.9% ± 1.9%. This good agreement corresponds to the smallest discrepancy between simulated and measured κ at this SS. Although κ was slightly overestimated at SS = 0.37%, the simulated and measured $D_{crit}$ remained very similar and both lay to the left of the PSD peak (Fig. S18b). As a result, the broader and flatter PSD did not introduce a noticeable bias in simulated CCN.

In contrast, at SS = 0.19%, the simulated CCN number concentrations were obviously overestimated by a factor

of >4 throughout the reaction. At this lowest SS, the required $D_{crit}$ is largest, and both simulated and measured $D_{crit}$ were located to the right of the PSD peak (Fig. S18a). The high bias in simulated $\kappa$ at this SS further reduced the simulated $D_{crit}$, and this underestimation, combined with the broader and flatter simulated PSD, resulted in pronounced overprediction of CCN relative to the measurements."

Figure S18 in the main text corresponds to Fig. R11 here.

[Figure]

**Fig. R11: (a-d) Measured and simulated number particle size distribution (dN/dlogDp) at four SS and corresponding measured or simulated critical activation dry diameter ($D_{crit}$).**

In addition to the above revisions as our response to the reviewer, we have also polished the obscure and poorly expressed sentences in the revised manuscript.

Abstract in line 18-33:

"**Abstract.** Secondary organic aerosols (SOA) contribute significantly to cloud condensation nuclei (CCN), which depend on particle size distribution (PSD), chemical composition and the hygroscopicity parameter ($\kappa$). Simulating SOA and CCN in chemical transport models relies on parameterizations, which need to be evaluated and improved against process-level models as a benchmark. Here, we simulated SOA concentration, chemical composition, PSD, $\kappa$, and CCN in $\alpha$-pinene ozonolysis, a classical system for SOA studies, using a process-level box model PyCHAM with near-explicit chemical mechanisms. We assessed how CCN, chemical composition, PSD and $\kappa$ can be modelled against measurements and evaluated the influence of these factors on CCN simulation. The model well simulated SOA mass concentration but overestimated O:C and H:C ratios, suggesting a possible lack of particle-phase chemistry. Highly oxygenated molecules (HOMs) contributed substantially to SOA mass. Simulated $\kappa$ closely agreed with measurements at moderate supersaturation (0.37%) but was overestimated at low supersaturation (0.19%) and underestimated at high supersaturation (0.55% and 0.73%). Particle growth and number concentrations were reasonably reproduced, though the simulated PSD was broader and flatter than measurement. Simulated CCN concentrations agreed well with measurements at moderate to high supersaturation (0.37–0.73%) but were overestimated at low supersaturation (0.19%). Sensitivity analysis highlights the importance of accurately representing both PSD and $\kappa$ for reliable CCN prediction, especially at supersaturation < 0.4%. This study also highlights that HOM formation, finer PSD resolution and improved $\kappa$ parameterizations are warranted in future chemical transport models, and evaluates the ability and limitations of this benchmark model."

**References**

Kreidenweis, S. M., Koehler, K., DeMott, P. J., Prenni, A. J., Carrico, C., and Ervens, B.: Water activity and activation diameters from hygroscopicity data - Part I: Theory and application to inorganic salts, Atmos. Chem. Phys., 5, 1357–1370, https://doi.org/10.5194/acp-5-1357-2005, 2005.

Moore, R., Nenes, A., and Medina, J.: Scanning Mobility CCN Analysis– A method for fast measurements of size resolved CCN distributions and activation kinetics, Aerosol Sci. Tech., 44, 861–871, DOI:10.1080/02786826.2010.498715, 2010.

O'Meara, S. P., Xu, S., Topping, D., Alfarra, M. R., Capes, G., Lowe, D., Shao, Y., and McFiggans, G.: PyCHAM (v2.1.1): a Python box model for simulating aerosol chambers, Geosci. Model Dev., 14, 675–702, https://doi.org/10.5194/gmd-14-675-2021, 2021.

Petters, M. D. and Kreidenweis, S. M.: A single parameter representation of hygroscopic growth and cloud condensation nucleus activity, Atmos. Chem. Phys., 7, 1961–1971, https://doi.org/10.5194/acp-7-1961-2007, 2007.

Roldin, P., Ehn, M., Kurtén, T. et al.: The role of highly oxygenated organic molecules in the Boreal aerosol-cloud-climate system, Nat Commun, 10, 4370, https://doi.org/10.1038/s41467-019-12338-8, 2019.

Zaveri, R., Easter, R., Fast, J., and Peters, L.: Model for Simulating Aerosol Interactions and Chemistry (MOSAIC), J. Geophys. Res., 113, D13204, https://doi.org/10.1029/2007JD008782, 2008.

Zhao, D. F., A. Buchholz, B. Kortner, P. Schlag, F. Rubach, A. Kiendler-Scharr, R. Tillmann, A. Wahner, J. M. Flores, Y. Rudich, et al.: Size-dependent hygroscopicity parameter (κ) and chemical composition of secondary organic cloud condensation nuclei, Geophys. Res. Lett., 42, 10,920–10,928, https://doi:10.1002/2015GL066497, 2015.

Zhao, D. F., Buchholz, A., Kortner, B., Schlag, P., Rubach, F., Fuchs, H., Kiendler-Scharr, A., Tillmann, R., Wahner, A., Watne, Å. K., Hallquist, M., Flores, J. M., Rudich, Y., Kristensen, K., Hansen, A. M. K., Glasius, M., Kourtchev, I., Kalberer, M., and Mentel, Th. F.: Cloud condensation nuclei activity, droplet growth kinetics, and hygroscopicity of biogenic and anthropogenic secondary organic aerosol (SOA), Atmos. Chem. Phys., 16, 1105–1121, https://doi.org/10.5194/acp-16-1105-2016, 2016.

Zhang, C., Guo, Y., Shen, H., Luo, H., Pullinen, I., Schmitt, S. H., et al.: Contrasting influence of nitrogen oxides on the cloud condensation nuclei activity of monoterpene-derived secondary organic aerosol in daytime and nighttime oxidation, Geophys. Res. Lett., 50, e2022GL102110, https://doi.org/10.1029/2022GL102110, 2023.

---

## Author Comment (AC2)

We thank the reviewer Simon Patrick O'Meara for the comments on our manuscript. The comments and suggestions are greatly appreciated. All the comments have been addressed and we believe that the revisions based on these comments have improved the quality of our manuscript. Below please find our responses to the comments one by one and the corresponding revisions made to the manuscript. The original comments are in italics. The revised parts of the manuscript are in blue here and can be followed in the revised manuscript with track changes with line numbers indicated.

Responses to reviewer 2:

*Reviewer: 2*

*Comments:*

*Song et al. 2025 provide an investigation into cloud condensation nuclei sensitivities that is novel in taking a near-explicit approach to gas- and particle-phase composition, allowing this composition to affect CCN. It is a meaningful investigation for the atmospheric science community, as it demonstrates the abilities and limitations of this bottom-up approach. An accurate version of such an approach would be valuable to the community to provide a benchmark against which parameterised approaches could be compared or even trained, therefore the work contributes toward realising such a standard. Given this novelty and significance, and that the method appears sound, I recommend publication in Atmospheric Chemistry and Physics after attention is paid to the suggested revisions below.*

*1. I recommend 'Explicit' in the title be changed to 'Near-explicit' if the gas-phase chemistry is being referred to. Potentially 'Process-level' would be more accurate since several processes are being considered.*

**Response:**

Accepted. This study employs the PyCHAM box model together with a near-explicit gas-phase chemical mechanism (MCM+PRAM) to simulate SOA mass, chemical composition, and size distribution, and—within our knowledge—is the first to extend these simulations to SOA-derived CCN activation. PyCHAM is one of the few open-source box models capable of treating comprehensive aerosol processes. In addition to gas-phase chemistry, it includes gas–particle partitioning, wall loss, nucleation, coagulation, and other microphysical processes, thereby allowing explicit simulation of both particle-phase chemical composition and size evolution.

Given that our analysis relies on these microphysical and chemical processes, and that the term near-explicit more specifically describes the gas-phase chemistry, we agree that process-level more accurately captures the scope of our work. Accordingly, we have revised the manuscript title to:

"Process-level simulation of chemical composition, size distribution and cloud condensation nuclei of secondary organic aerosol from α-pinene ozonolysis"

*2. Line 29 and elsewhere, please check grammar in phrases like 'exhibited wider and flatter size distribution' so that they are grammatically correct, e.g.: 'exhibited a wider and flatter size distribution' or 'exhibited wider and flatter size distributions'*

**Response:**

Accepted. We have revised the sentence as follows (line 27-28). Additionally, we have corrected the grammatical

errors in the revised manuscript and further polished the language throughout the paper.

"Particle growth and number concentrations were reasonably reproduced, though the simulated PSD was broader and flatter than measurement."

*3.    Line 55 and surrounding text. Whilst the authors explain the limitations of 3D models to identify CCN sensitivities (perhaps too much given that it is well known that such models depend on higher physicochemical resolution experiments and models to develop their parameterisations), the workflow between the presented work and the 3D models is not well explained, leaving the reader with some uncertainty about the significance of the current work. I think I understand this workflow, and therefore why the work is important, and have included my thoughts in my opening paragraph of this review. I recommend that the authors be less critical of 3D models and more constructive in describing why work like theirs can, eventually (once a sufficiently accurate model is verified), support the 3D model development. I think the revised messaging in this section of the introduction also needs to be folded into the summary of the study in the final paragraph of the introduction, where the aims of the study are stated. The current aims of the study are not very significant at all, as past studies have identified key components in the alpha-pinene system (such as Roldin et al. 2019), and others have identified key sensitivities in CCN evolution (such as McFiggans et al. 2006 (doi.org/5194/acp-6-2593-2006), if this last article is not yet cited in the introduction I think it, or a more recent article that updates its findings, should be cited)*

**Response:**

We thank the reviewer for the constructive advice which helps streamline the significance of our study. The ultimate goal of this work is indeed to reduce CCN simulation uncertainties and to provide guidance for improving 3D model treatments, which was not clearly reflected in the original introduction. Following the reviewer's suggestion, we have revised the introduction to reduce the discussion of limitations of 3D models and to clarify the logical flow as follows (line 43-62).

"Therefore, uncertainties in modeled CCN levels are strongly influenced by these parameters (McFiggans et al., 2006). In current chemical transport models, to enhance computational efficiency and numerical stability, SOA and CCN formation relies on simplified parameterizations which have been developed and optimized based on laboratory measurements or ambient data (Hodzic and Jimenez, 2011). For example, lumped species and reactions are usually adopted for gas-phase chemical mechanisms. Limited aerosol size bin resolution is typically used to represent size distribution and number concentration (Kanakidou et al., 2005; Yu and Luo, 2009; Luo and Yu, 2011; Topping and Bane, 2022). The volatility bases set (VBS) and its derivatives are often used to represent chemical composition of SOA via gas-particle partitioning (Donahue et al., 2006). Moreover, the hygroscopicity parameter ($\kappa$), derived from Köhler theory (Petters and Kreidenweis, 2007), is parameterized either as a uniform value for organic aerosols (OA) in most global models (Fanourgakis et al., 2019) or as several constant values for different OA types in regional models (Wang et al., 2019; Kuang et al., 2020). While these parameterizations or simplifications provide useful and efficient approaches to model SOA composition, concentrations, and CCN concentrations, it is necessary to evaluate them against process-level models as a benchmark. Such process-level models can provide a mechanistic representation of SOA chemistry and corresponding CCN formation by incorporating explicit or near-explicit chemical mechanisms and physicochemical processes such as detailed chemistry, gas-particle partitioning, fine particle size bin, and explicit treatment of $\kappa$. Such

models are suitable for simulating SOA and CCN in chamber or laboratory studies and for developing more detailed bottom-up parameterizations applicable to chemical transport models. Moreover, such models can be used to improve chemical transport models, potentially through training artificial intelligence (AI)-based models capable of learning detailed parameterizations (Xia et al., 2025). In addition, this process-level approach enables the assessment of factors controlling SOA-derived CCN based on explicit chemical composition (which determines κ) and PSD simulations."

Regarding possible chemical processes in the α-pinene ozonolysis system, since our chemical mechanism is consistent with Roldin et al. (2019), we will not position mechanism exploration as a main research objective. Nonetheless, in the Results section we will still discuss chemical pathways that may influence SOA formation and composition.

The previous statement- "highlights the importance of accurate simulation of κ and particle size distribution in CCN simulations"- was indeed too general, as prior theoretical, experimental, and modeling studies have already identified the key aerosol properties affecting CCN activation. McFiggans et al. (2006), for example, synthesized the major controlling factors, including particle size, composition, and mixing state (already cited in the manuscript). Our study specifically focuses on how the model representation of SOA size distribution and hygroscopicity influences CCN predictions. More precisely, our objective is to use a near-explicit, process-level approach to investigate how SOA-derived CCN respond to key controlling factors, thereby demonstrating how this methodology can evaluate and inform the development of CCN parameterizations in 3D models. In the revised manuscript we have underlined the purpose and significance of this study (line 102-110).

"In this study, we simulated the concentration and chemical composition of SOA formed from α-pinene ozonolysis, a benchmark system in SOA studies, and CCN concentrations using PyCHAM model. Simulated SOA mass, number concentrations, chemical composition, size distribution, κ, and CCN number concentrations were evaluated against measurements. We further investigated the impact of SOA κ and PSD representation on CCN. This study aims to evaluate the capability and limitations of process-level modeling of SOA concentrations, chemical composition, PSD, and CCN using a bottom-up approach as a potential benchmark model. In the future, once validated and improved such a benchmark model for SOA and CCN simulation can be used for assessing SOA and CCN parameterizations in chemical transport models and may be used to improve chemical transport models, potentially through training AI-based models capable of learning detailed parameterizations."

4. *Line 60 and elsewhere, when referring to the chemistry, please use 'near-explicit' rather than 'explicit' to help clarify the type of mechanism applied (MCM papers often use the 'near-explicit' description)*

**Response:**

Accepted. We have revised the manuscript to replace references to an "explicit" chemical mechanism with the more accurate term "near-explicit" in line 55-56.

"…by incorporating explicit or near-explicit chemical mechanisms and physicochemical processes such as detailed chemistry…"

And line 63:

"Over the past two decades, based on comprehensive explicit or near-explicit gas-phase chemical mechanisms…"

And line 68-69:

"…the most widely used near-explicit mechanism (Jenkin et al., 1997, 2003; Saunders et al., 2003). Similar near-explicit or explicit chemical mechanisms include…"

And other relevant locations.

5.    *Lines 75-76 the dynamic transfer mentioned is based on thermodynamic absorption partitioning theory, so I'm not sure there is a distinction between the approaches as currently named (to justify the 'or' in this sentence). If the authors want to distinguish between equilibrium partitioning and dynamic partitioning, please make this distinction clearer.*

**Response:**

Accepted. We agree that dynamic gas-particle mass transfer is based on thermodynamic absorption partitioning theory. We have updated the manuscript to make the distinction between these two partitioning approaches clearer (line 77-79).

"In the models simulating SOA formation, gas-particle partitioning has been modeled based on thermodynamic absorption equilibrium partitioning theory (Pankow, 1994) or dynamic gas-particle mass transfer partitioning (Seinfeld and Pandis, 2016)."

6.    *Line 76, I think a more balanced introduction to the importance of particle-phase reactions can be provided, and would be helpful for the reader, for example Lopez et al. (2025) doi.org/1039/d5ea00062a suggest that for the alpha-pinene ozonolysis system at relatively low temperatures, particle-phase reactions play a minor role in SOA evolution.*

**Response:**

Accepted. Lopez et al. (2025) developed the diagonal VBS (dVBS) method, which incorporates both gas-phase and particle-phase observations to distinguish between growth driven by gas-phase precursor condensation and growth driven by particle-phase chemical reactions. In their low-temperature α-pinene ozonolysis experiments, they concluded that particle growth was dominated by gas-phase precursor condensation, with particle-phase reactions playing a relatively minor role. In the revised manuscript, we have discussed the importance of particle-phase reaction in a more-balanced way as follows (line 79-81).

"Besides gas-phase reaction and gas-particle partitioning, particle-phase reactions, such as oligomerization and polymerization, have also been shown to affect SOA composition in the model simulation, although their importance varies across environmental conditions and reaction systems…"

7.    *Line 86, when discussing the simulation of SOA composition I think the articles of Roldin et al. 2019 (PRAM paper) and Pichelstorfer et al. (2024) (autoAPRAM-fw) paper deserve to be mentioned as they reflect advances in HOMs simulations with comparisons against CIMS observations.*

**Response:**

Accepted. We have added further discussion on the current status of SOA chemical composition modeling, with particular emphasis on the simulation of HOMs (Roldin et al., 2019; Pichelstorfer et al., 2024), in line 92-95 in the revised manuscript.

"Roldin et al. (2019) reproduces SOA mass from α-pinene ozonolysis based on reasonable HOM simulation but

overestimates H:C, while O:C shows smaller bias at the average level. Pichelstorfer et al. (2024) captures the mass distribution of gas-phase HOMs and the monomer/dimer ratio, and achieves good agreement with SOA yield under low-$NO_x$ conditions, but underestimates SOA formation under high-$NO_x$ conditions.”

8. *Aims in final paragraph of introduction, please see my point three above.*

**Response:**

Accepted. We have reorganized the narrative flow and revised the statement of our research aims accordingly. The specific revision can be found in the response to comment 3.

9. *Section 2.2: the version of PyCHAM used should be stated (provided in the setup file of PyCHAM), and the relevant input files used in the study should be provided via a URL in the data availability section, with a reference to this section provided in the main text. Figure S2 indicates that best agreement with [SOA] is gained when the walls do not absorb gases. The main text should justify why a Cw>0 is therefore used and whether the chosen Cw>0 is physically realistic.*

**Response:**

We have added the PyCHAM version used (5.5.9) in Sect. 2.2, as well as a reference to the input files provided in the Data Availability section in line 143-144.

“The α-pinene ozonolysis experiment was simulated using the PyCHAM (CHemistry with Aerosol Microphysics in Python) model (v5.5.9) (O'Meara et al., 2021).”

And line 235-236:

“And the input files including model variables setting and chemical mechanism files used in PyCHAM are supplied in Sect. Data availability.”

And line 507-508:

“The input files including model variables setting and chemical mechanism files used in PyCHAM are available on Zenodo at https://doi.org/10.5281/zenodo.17539325.”

During chamber experiments with Teflon walls, a positive effective wall mass concentration ($C_w>0$) means that gas-phase organics can be absorbed in the chamber wall. The absorption of gas-phase organics into Teflon chamber walls have been reported by a number of studies. Matsunaga and Ziemann (2010) and Zhang et al. (2014) systematically quantified gas-wall partitioning of organic compounds in Teflon chambers. Neglecting gas-wall loss would lead to an overestimation of both gaseous products and SOA yields, highlighting that gas-wall partitioning is a non-negligible process in chamber studies.

Our simulations were designed on the basis of the experimental conditions and aimed to reproduce the chamber processes as realistically as possible. As shown in Fig. S2, when $C_w = 0$, the simulated SOA mass concentration is still underestimated, which indicates that the source of SOA mass is underestimated in current model setup. The result for $C_w = 1 \times 10^{-10}$ g m$^{-3}$ is nearly identical to that for $C_w = 0$. However, to reflect the physical relevance of wall absorption in chamber studies, we use $C_w = 1 \times 10^{-6}$ g m$^{-3}$ in our simulations. When $C_w$ increases by one order of magnitude from $1 \times 10^{-6}$ g m$^{-3}$ to $1 \times 10^{-5}$ g m$^{-3}$, the deviation in simulated SOA mass concentration becomes substantially larger. In contrast, reducing $C_w$ by one order of magnitude ($1 \times 10^{-7}$ g m$^{-3}$) produces only minor changes in SOA mass concentration.

Therefore, $C_w = 1 \times 10^{-6}$ is selected for this study.

We have revised the relevant explanation of $C_w$ in the main text (line 172-182).

"Meanwhile, $C_w$ (g m$^{-3}$) represents wall adsorption/absorption properties, including effects of RH, surface area, diffusivity, and porosity. Neglecting the gas-wall partitioning of organic compounds in Teflon film chambers in the model can lead to a systematic overestimation of the yields of gaseous products and SOA (Matsunaga and Ziemann, 2010; Zhang et al., 2014). Therefore, we conducted a sensitivity analysis of SOA mass concentration by testing several different orders of magnitude for the $C_w$ value (Fig. S2). When $C_w = 0$, the simulated SOA mass concentration was still underestimated, which indicated that the source of SOA mass was underestimated in current model setup. The result for $C_w = 1 \times 10^{-10}$ g m$^{-3}$ was nearly identical to that for $C_w = 0$. However, to reflect the physical relevance of wall absorption in chamber studies, we used $C_w = 1 \times 10^{-6}$ g m$^{-3}$ in our simulations, and the simulated SOA mass was reasonably reproduced. Increasing $C_w$ by one order of magnitude to $1 \times 10^{-5}$ g m$^{-3}$ resulted in a larger deviation in SOA mass concentration, whereas decreasing it by one order of magnitude to $1 \times 10^{-7}$ g m$^{-3}$ led to only minimal changes in SOA mass concentration. Consequently, $C_w = 1 \times 10^{-6}$ g m$^{-3}$ was finally selected for this study."

*10. Around line 170 and its paragraphs, I agree with the other reviewer that more detailed is needed about how the two different approaches to applying PyCHAM influence our interpretation of the results: what conclusions can we still make with this approach and what conclusions are no longer available (but would have been if the same approach was used throughout)*

**Response:**

We initially applied the nucleation scheme to simulate SOA mass and composition. However, because nucleation is controlled by only three user-defined parameters and lacks a physically justified description of the early particle size distribution (PSD), the simulated PSD deviates substantially from observations, which affects CCN predictions. To improve PSD, we also tested a seed scheme by constraining early growth with SMPS-measured PSD and assuming $C_{20}H_{30}O_{17}$ as the seed. This yields a better PSD (Fig. R1) but artificially increases SOA O:C toward that of the assumed seed (Fig. R2 and Fig. S6 in the Supplement), ultimately producing poorer agreement with observations than the nucleation scheme.

[Figure]

**Fig. R1: Measured and simulated number particle size distribution (dN/dlogDp) by the two schemes (nucleation or seed) at the reaction time of 2 h.**

[Figure]

**Fig. R2: O:C and H:C distributions of SOA measured experimentally and simulated using the two schemes (nucleation and seed).**

To obtain accurate CCN, we therefore used a combined approach: SOA κ from the nucleation scheme and PSD from the seed scheme. We acknowledge that this may reduce generality. To assess the influence of each scheme, we first applied the nucleation scheme consistently for both κ and CCN (Fig. R3). The resulting CCN are similar to those of the combined scheme at SS = 0.55% and 0.73%, whereas at SS = 0.19% they exhibit a trend of initial increase followed by a decrease, and at SS = 0.37% they are overestimated. We then applied the seed scheme consistently for κ (Fig. R4) and CCN (Fig. R5). CCN predictions remain comparable to that of combined approach across all SS, though slightly lower due to reduced κ.

Overall, if the nucleation scheme is applied alone, the simulated PSD performs worse than that obtained with the combined approach, resulting in larger bias of CCN concentrations at the two lower SS. In contrast, applying the seed scheme alone leads to worse simulations of initial SOA mass concentration, chemical composition, and κ due to the assumed composition of seed species. However, because the PSD remains relatively accurate, the resulting CCN concentrations are similar to those from the combined approach. Therefore, in this study we adopted the combined approach, which reconciles the simulations of both chemical composition and PSD while minimizing bias in CCN predictions.

[Figure]

**Fig. R3: CCN number concentrations (# cm⁻³) measured experimentally and simulated using κ and PSD from nucleation scheme.**

[Figure]

**Fig. R4: Measured and simulated (using the seed scheme) SOA κ.**

[Figure]

**Fig. R5: Same as Fig. R3, but for CCN simulated by κ and PSD from seed scheme.**

We have added the description about the impact of two independent schemes on CCN predictions in Sect. 3.4 (line 452-465).

"**3.4 Discussion of the influence of individual schemes (nucleation vs. seed) on CCN predictions**

To demonstrate the rationale for the combined approach - using κ from the nucleation scheme together with PSD from the seed scheme - a detailed analysis of the effect of applying each scheme independently on the CCN simulations is implemented.

As shown in Fig. S21, CCN calculated using the κ by the nucleation scheme (Fig. 6) and PSD by the same scheme (Fig. S4) at SS = 0.55% and 0.73% were comparable to those from the combined-scheme approach. However, at SS = 0.37%, CCN was moderately overestimated, and at SS = 0.19% the predicted CCN was initially higher than the measurements and then decreased toward zero. In contrast, CCN calculated using the κ from the seed scheme (Fig. S22) combined with its PSD (Fig. 5) produced lower CCN across all four SS (Fig. S23), leading to a worse performance than that of the combined-scheme approach.

Overall, if the nucleation scheme was applied alone, the simulated PSD performed worse than that obtained with the combined approach, resulting in deviations of CCN concentrations at the two lower SS. In contrast, applying the

seed scheme alone led to worse simulations of initial SOA mass concentration, chemical composition, and κ due to the assumed composition of seed species. However, because the PSD remained relatively accurate, the resulting CCN concentrations were similar to those from the combined approach."

Fig. S21-23 correspond to Fig. R3-5 here.

*11. Line 190, given that particle sizes vary by a factor of 10 in this study, and that particle losses to wall are sensitive to size (e.g. data contained for chambers here https://www.eurochamp.org/simulation-chambers), there either needs to better justification for using a size independent Beta_flec (e.g. a supplementary material showing size independence across particle sizes), or a size-dependent particle loss rate to walls needs to be used.*

**Response:**

We agree that particle wall loss rate depends on particle size. However, in our study, in the simulation period when PSD is not constrained by measurements (after 0.6 h), the size range varies only by a factor of ~2. We determined the value of $\beta$ based on the observed decay slope of SOA mass concentration during the final hour of the experiment. We also found that, after the experiment ended, the SMPS number PSD exhibited no significant shift in peak diameter over time; that is, the size distributions during this period were very similar, with only minor differences at the lower and upper size limits (Fig. R6). This behavior indicates that the particle wall-loss rate did not exhibit strong size dependence within the particle size range of the simulation period.

We have added the description of this issue in the revised manuscript as follows (line 223-226).

"As the PSD size range varied by only a factor of ~2 during the period when it was not constrained by SMPS measurements, and no obvious deviation in peak diameter was observed within ~1 h after the experiment ended, this study used a uniformed value ($\beta_{flec} = 2.37\times10^{-5}$ s$^{-1}$) based on the measured particle loss rates without considering the size dependence."

[Figure]

**Fig. R6: The time evolution of dN/dlogDp measured by SMPS within 1 h after the experiment ends.**

*12.  Line 245, I don't think the evidence presented up to this point justifies the comment about likely attribution, and possible attribution seems more fitting at this point. Also around this line, the authors mention the activity coefficient. Could there be some more discussion, with reference to the literature, about whether a higher solubility (represented by the activity coefficient) of the partitioning components could bridge the gap to observed SOA. In Section 2.2 please state the values used for activity coefficients and accommodation coefficients in simulations.*

**Response:**

Accepted. We agree with the reviewer that "likely" should be replaced with "possibly" to maintain a more accurate tone as other factors such as non-ideality and uncertainty in HOMs simulation can also contribute to the underestimation of SOA mass, and we have made this revision in the revised manuscript (line 294-295).

"Therefore, the underestimation of SOA mass is possibly attributed to missing particle-phase chemistry."

We have also expanded our discussion of gas-phase products and HOMs.

Regarding the activity coefficient, as emphasized by Zuend and Seinfeld (2012), accounting for non-ideality is crucial for accurately simulating SOA mass yields and O:C. Lannuque et al. (2023) likewise incorporated non-ideal interactions between organic molecules and inorganic ions in the aqueous phase when modeling SOA partitioning, and showed that assuming ideal behavior can lead to substantial underestimation of SOA formation—especially in systems lacking a pre-existing organic phase. Their results demonstrate that considering both organic- and aqueous-phase non-ideality is necessary for realistic predictions of SOA formation and composition. As indicated in Equation (2) of the manuscript, a lower γ (corresponding to a higher solubility) would theoretically lead to a higher concentration of the species in the particle phase, thereby increasing SOA mass concentration.

Because no well-established data of activity coefficient γ were available for our experimental conditions, we only simulated the idealized conditions. Non-ideality was neglected, and all activity coefficient γ were set to 1. In our simulations, the accommodation coefficients α for all species were assumed to be 1. These parameter choices have been added explicitly to the revised manuscript (line 155-156).

"Because no well-established data of γ were available for our experimental conditions, we only simulated the idealized conditions (i.e. γ for all components were set to 1)."

And line 162:

"In our simulation, $\alpha_i$ for all components were set to 1."

And we have added more discussion on activity coefficient in the revised manuscript as follows (line 282-293).

"The gas-particle partitioning in the model, which can be adjusted by activity coefficient (γ), also influences SOA mass concentration. The impact of non-ideal behavior on simulated gas-particle partitioning of SOA has been explored in previous studies (Zuend and Seinfeld, 2012; Lannuque et al., 2023). For example, Zuend and Seinfeld (2012) improved the accuracy of simulated mass and composition of SOA formed from α-pinene ozonolysis by accounting for non-ideal mixing and liquid–liquid phase separation through the calculation of γ for components in the liquid mixture using AIOMFAC (Aerosol Inorganic-Organic Mixtures Functional groups Activity Coefficients). Lannuque et al. (2023) also accounted for non-ideality (i.e., interactions between organic molecules and inorganic ions in the aqueous phase) in their simulation of SOA gas-particle partitioning. They found that considering only ideal partitioning leads to a substantial underestimation of SOA formation, particularly in the absence of a pre-existing organic phase. As no well-established

data of $\gamma$ were available for our experimental conditions, we only simulated the idealized conditions. Non-ideality was neglected, and $\gamma$ for all species were set to 1. As shown in Equation (2), lower $\gamma$ (corresponding to higher solubility) would lead to higher SOA concentrations and thus reduce the gap between simulated and measured SOA mass concentrations."

*13.  Line 267, again I think 'likely attributed' is not yet supported by this stage of the evidence. 'Possibly attributed' at this stage would be more suitable.*

**Response:**

Accepted. We have revised the wording to "possibly attributed" at the corresponding location in the revised manuscript (line 320-321).

"The overestimation (32.4% ± 2.2%) of O:C is possibly attributed to the absence of particle-phase reactions in our simulations as mentioned above…"

*14.  Line 281, rather than 'alter', can you tell the reader which direction the particle-phase processes mentioned in the referenced works take the H:C and O:C values.*

**Response:**

Accepted. We have made the description more specific regarding how particle-phase chemistry influences SOA chemical composition, based on findings from previous studies as follows (line 332-336).

"Oxidation of particle-phase organics by atmospheric oxidants typically leads to an increase in O:C of organic matter by functionalization introducing oxygen-containing functional groups like -OH, -COOH, -ONO$_2$, -OOH, or leads to a decrease in O:C by fragmentation i.e. C-C bond breaking or peroxide photolysis (Kroll and Seinfeld, 2008; Hallquist et al., 2009), and their absence in simulations possibly contributes to the discrepancy between modelled and measured time series of O:C and corresponding H:C ratios."

*15.  Around line 240, could the authors provide some more quantification of the monomer:dimer ratio (observed and simulated). Comparing this ratio between observation and simulation, could the authors please, discuss, e.g. around where O:C and H:C are discussed, whether inaccuracies in the gas-phase chemistry could explain the O:C and H:C discrepancy?*

**Response:**

We originally compared simulated and observed gas-phase HOMs by normalizing both mass spectra to their respective maximum signal and evaluating the overall spectral pattern. As the reviewer noted, this approach provides limited quantitative support for assessing model performance. We have therefore replaced this analysis with a direct comparison of the simulated and observed concentrations of gas-phase HOMs (Fig. R7). Although the simulated total concentration of gas-phase HOMs during the first 5 min of the experiment (0.011 ppb) is slightly underestimated compared to the measurement (0.014 ppb), the simulation reproduces the observed HOMs species (m/z) reasonably well. Specifically, the concentration levels of dimers are captured closely, while those of monomers are underestimated, particularly at m/z < 300. Our spectra pattern is similar to the findings of Roldin et al. (2019), especially for dimers, who also showed a slight underestimation of monomers. Furthermore, the fractions of HOMs monomers and dimers are also

well captured (Fig. R8). These findings suggest that the gas-phase chemical mechanism employed in the model is generally reasonable. In addition, the simulated O:C of gas-phase HOMs within the first 10 min agrees well with measurement, whereas the H:C is overestimated (Fig. R9). This suggests that gas-phase chemistry alone cannot account for the discrepancy in the O:C of SOA, although it may contribute to the overestimation of the H:C in SOA. Given the generally reasonable performance of gas-phase chemistry and gas-particle partitioning, we attributed the discrepancies in SOA mass concentration and O:C and H:C possibly to the absence of particle-phase chemistry in the model.

[Figure]

**Fig. R7: Measured and simulated gas-phase HOMs mass spectra averaged over the first 5 min of experiment, during which gas-phase HOMs were rapidly accumulated and particle-phase concentrations were low.**

[Figure]

**Fig. R8: Pie charts of (a) measured and (b) simulated gas-phase HOMs monomer and dimer fractions averaged over the first 5 min of the reaction.**

[Figure]

**Fig. R9: O:C and H:C ratios for measured (circles) and simulated (squares) gas-phase HOMs within the first 10 min of the experiment.**

We have added a more detailed discussion on this in the revised manuscript (line 276-280).

"The gas-phase chemistry, including the loss of α-pinene (Fig. S7) and the composition of HOMs, is generally well reproduced (Fig. S9). The model reproduces the bimodal distributions of HOM monomers (m/z 230-380) and dimers (m/z 400-550), although the concentration of monomers is underestimated, especially below m/z 300. It also reasonably captures the fractions of HOM monomers and dimers (Fig. S10), while showing a slight underestimation of dimers in the simulation."

Fig. S9-10 correspond to Fig. R7-8 here.

And line 317-322:

"Figure S11 illustrates that the simulated O:C of gas-phase HOMs in the first 10 min of reaction is consistent with measurements, while the H:C is moderately overestimated. These results imply that inaccuracies in the simulated gas-phase chemistry may contribute to the overestimation (21.2% ± 2.1%) of the H:C in SOA, but exert only a minor influence on the O:C in SOA. The overestimation (32.4% ± 2.2%) of O:C is possibly attributed to the absence of particle-phase reactions in our simulations as mentioned above, emphasizing the importance of particle-phase chemistry in determining SOA chemical composition."

Figure S11 corresponds to Fig. R9 here.

*16. Results and Conclusion section. Given that studies like this are necessary to assess how helpful process-level modelling currently is to informing 3D model parameterisations, there should at least be a discussion around what the results mean to this workflow. E.g., is all, or some parts, of the simulation sufficiently accurate as to be used to inform 3D models? Where should future work be focused to close any remaining gaps before we can reliably apply these process*

*models to gain 3D model parameterisations?*

**Response:**

Accepted. As mentioned in response to comment 3, our aim is to simulate SOA and CCN using a near-explicit, process-level model and verify the potential of this methodology to inform the development of parameterizations in 3D models.

First, because the simulations are conducted under controlled chamber conditions with corresponding observational constraints, we are able to directly and transparently assess how CCN formation responds to changes in SOA size or composition. This makes the methodological framework both reasonable and convincing.

However, the accuracy of the simulated results is still limited by the current capabilities of the PyCHAM model. For example, particle-phase chemistry cannot yet be incorporated; activity coefficients are not explicitly treated; and the model lacks physically constrained descriptions of nucleation and initial size distributions. Improvements in these aspects are warranted for future model development.

Despite these limitations, SOA mass, chemical composition, PSD, and CCN simulations show good agreement with measurements, which supports the reliability of this process-level approach. And the lack of HOMs gas-phase chemistry leads to deviation of SOA simulation. CCN sensitivity analysis further indicates that 8 size bins of PSD and uniformed κ of 0.1 are not enough to appropriately describe CCN number concentration. These results inform 3D models that HOMs chemistry, more detailed size bin resolution, and κ are warranted in future work.

Moreover, future work on model development such as adding particle-phase chemistry, treating activity coefficients and describing nucleation with initial size distribution are also warranted.

And future work should extend this analysis to additional VOC oxidation systems, in order to develop generalized parameterizations of particle size and κ that are applicable for 3D models, possibly by training an AI model which has the potential to learn the sophisticated parameterizations.

In the revised manuscript, we have added the corresponding discussions as follows (line 474-484).

"This study advances previous research by simulating CCN formation from SOA using a near-explicit and process-level model for the first time to our knowledge, and demonstrates the potential of this approach to inform the development of parameterizations in chemical transport models. Although current model still has some limitations, such as the absence of particle-phase chemistry, lack of explicit treatment of activity coefficients, and the inability to prescribe physically based nucleation and initial PSD, the model reproduces key features of the measured SOA and CCN reasonably well. Our findings further indicate that simplified representations of PSD and κ are insufficient for accurately describing CCN. These results suggest that HOMs chemistry, finer PSD resolution and improved κ parameterizations in chemical transport models are warranted. Improvement in process-level model e.g. including particle-phase chemistry, explicit treatment of activity coefficients, and allowing prescribing physically based nucleation and initial PSD are warranted in future work to provide a benchmark model to evaluate various parameterizations related to SOA formation and CCN concentrations. And future work could also extend this analysis to a range of biogenic and anthropogenic VOCs oxidation systems to develop generalized PSD and κ parameterization modules, potentially through training AI-based methods capable of learning sophisticated and process-informed parameterizations, which may be used to improve chemical transport models."

*17. Figure S6, in the y-axis title is the word normalized. The method for normalising both observation and simulation results needs to be stated.*

**Response:**

In the original manuscript, the gas-phase HOMs spectra were normalized to their respective maximum signal. To allow a more detailed comparison between the simulated and observed HOMs mass-spectral distributions, we have revised Fig. S6 (now it is Fig. S9) and presented the concentration (ppbv) in mass spectra directly in the revised manuscript. The specific revision and description can be found in response to comment 15.

In addition to the above revisions as our response to the reviewer, we have also polished the obscure and poorly expressed sentences in the revised manuscript.

Abstract in line 18-33:

"**Abstract.** Secondary organic aerosols (SOA) contribute significantly to cloud condensation nuclei (CCN), which depend on particle size distribution (PSD), chemical composition and the hygroscopicity parameter ($\kappa$). Simulating SOA and CCN in chemical transport models relies on parameterizations, which need to be evaluated and improved against process-level models as a benchmark. Here, we simulated SOA concentration, chemical composition, PSD, $\kappa$, and CCN in $\alpha$-pinene ozonolysis, a classical system for SOA studies, using a process-level box model PyCHAM with near-explicit chemical mechanisms. We assessed how CCN, chemical composition, PSD and $\kappa$ can be modelled against measurements and evaluated the influence of these factors on CCN simulation. The model well simulated SOA mass concentration but overestimated O:C and H:C ratios, suggesting a possible lack of particle-phase chemistry. Highly oxygenated molecules (HOMs) contributed substantially to SOA mass. Simulated $\kappa$ closely agreed with measurements at moderate supersaturation (0.37%) but was overestimated at low supersaturation (0.19%) and underestimated at high supersaturation (0.55% and 0.73%). Particle growth and number concentrations were reasonably reproduced, though the simulated PSD was broader and flatter than measurement. Simulated CCN concentrations agreed well with measurements at moderate to high supersaturation (0.37–0.73%) but were overestimated at low supersaturation (0.19%). Sensitivity analysis highlights the importance of accurately representing both PSD and $\kappa$ for reliable CCN prediction, especially at supersaturation < 0.4%. This study also highlights that HOM formation, finer PSD resolution and improved $\kappa$ parameterizations are warranted in future chemical transport models, and evaluates the ability and limitations of this benchmark model."

**References**

Kreidenweis, S. M., Koehler, K., DeMott, P. J., Prenni, A. J., Carrico, C., and Ervens, B.: Water activity and activation diameters from hygroscopicity data - Part I: Theory and application to inorganic salts, Atmos. Chem. Phys., 5, 1357–1370, https://doi.org/10.5194/acp-5-1357-2005, 2005.

Lannuque, V., D'Anna, B., Kostenidou, E., Couvidat, F., Martinez-Valiente, A., Eichler, P., Wisthaler, A., Müller, M., Temime-Roussel, B., Valorso, R., and Sartelet, K.: Gas–particle partitioning of toluene oxidation products: an experimental and modeling study, Atmos. Chem. Phys., 23, 15537–15560, https://doi.org/10.5194/acp-23-15537-2023, 2023.

Lopez, B., Nirvan Bhattacharyya, Jenna DeVivo, Mingyi Wang, Lucia Caudillo-Plath, Mihnea Surdu, Federico Bianchi, Zoé Brasseur, Angela Buchholz, Dexian Chen, Jonathan Duplissy, Xu-Cheng He, Victoria Hofbauer, Naser Mahfouz, Vladimir Makhmutov, Ruby Marten, Bernhard Mentler, Maxim Philippov, Meredith Schervish, Dongyu S. Wang, Stefan K. Weber, André Welti, Imad El Haddad, Katrianne Lehtipalo, Markku Kulmala, Douglas Worsnop, Jasper Kirkby, Roy L. Mauldin, Dominik Stolzenburg, Siegfried Schobesberger, Richard Flagan and Neil M. Donahue.: A diagonal volatility basis set to assess the condensation of organic vapors onto particles, Environ. Sci.: Atmos., 5, 1035-1061, https://doi.org/10.1039/D5EA00062A, 2025.

Matsunaga, A., and Ziemann, P. J. Gas-wall partitioning of organic compounds in a teflon film chamber and potential effects on reaction product and aerosol yield measurements. Aerosol Science and Technology, 44(10), 881–892. https://doi.org/10.1080/02786826.2010.501044, 2010.

McFiggans, G., Artaxo, P., Baltensperger, U., Coe, H., Facchini, M. C., Feingold, G., Fuzzi, S., Gysel, M., Laaksonen, A., Lohmann, U., Mentel, T. F., Murphy, D. M., O'Dowd, C. D., Snider, J. R., and Weingartner, E.: The effect of physical and chemical aerosol properties on warm cloud droplet activation, Atmos. Chem. Phys., 6, 2593–2649, https://doi.org/10.5194/acp-6-2593-2006, 2006.

O'Meara, S. P., Xu, S., Topping, D., Alfarra, M. R., Capes, G., Lowe, D., Shao, Y., and McFiggans, G.: PyCHAM (v2.1.1): a Python box model for simulating aerosol chambers, Geosci. Model Dev., 14, 675–702, https://doi.org/10.5194/gmd-14-675-2021, 2021.

Petters, M. D. and Kreidenweis, S. M.: A single parameter representation of hygroscopic growth and cloud condensation nucleus activity, Atmos. Chem. Phys., 7, 1961–1971, https://doi.org/10.5194/acp-7-1961-2007, 2007.

Pichelstorfer, L., Pontus Roldin, Matti Rissanen, Noora Hyttinen, Olga Garmash, Carlton Xavier, Putian Zhou, Petri Clusius, Benjamin Foreback, Thomas Golin Almeida, Chenjuan Deng, Metin Baykara, Theo Kurten and Michael Boy.: Towards automated inclusion of autoxidation chemistry in models: from precursors to atmospheric implications, Environ. Sci.: Atmos., 4, 879-896, https://doi.org/10.1039/D4EA00054D, 2024.

Roldin, P., Ehn, M., Kurtén, T. et al.: The role of highly oxygenated organic molecules in the Boreal aerosol-cloud-climate system, Nat Commun, 10, 4370, https://doi.org/10.1038/s41467-019-12338-8, 2019.

Zhang, X., Cappa, C.D., Jathar, S.H., McVay, R.C., Ensberg, J.J., Kleeman, M.J., and Seinfeld, J.H. Influence of vapor wall loss in laboratory chambers on yields of secondary organic aerosol, Proc. Natl. Acad. Sci. U.S.A. 111 (16) 5802-5807, https://doi.org/10.1073/pnas.1404727111, 2014.

Zuend, A. and Seinfeld, J. H.: Modeling the gas-particle partitioning of secondary organic aerosol: the importance of liquid-liquid phase separation, Atmos. Chem. Phys., 12, 3857–3882, https://doi.org/10.5194/acp-12-3857-2012,

2012.